# A neural implementation model of feedback-based motor learning

Barbara Feulner [1], Matthew G. Perich [2,3], Lee E. Miller [4,5,6], Claudia Clopath [1,7] ✉ & Juan A. Gallego [1,7] ✉

Animals use feedback to rapidly correct ongoing movements in the presence of a perturbation. Repeated exposure to a predictable perturbation leads to behavioural adaptation that compensates for its effects. Here, we tested the hypothesis that all the processes necessary for motor adaptation may emerge as properties of a controller that adaptively updates its policy. We trained a recurrent neural network to control its own output through an error-based feedback signal, which allowed it to rapidly counteract external perturbations. Implementing a biologically plausible plasticity rule based on this same feedback signal enabled the network to learn to compensate for persistent perturbations through a trial-by-trial process. The network activity changes during learning matched those from populations of neurons from monkey primary motor cortex − known to mediate both movement correction and motor adaptation − during the same task. Furthermore, our model natively reproduced several key aspects of behavioural studies in humans and monkeys. Thus, key features of trial-by-trial motor adaptation can arise from the internal properties of a recurrent neural circuit that adaptively controls its output based on ongoing feedback.

Animals, including humans, have a remarkable ability to rapidly correct their ongoing movements based on perceived errors even when feedback is distorted, such as when reaching into a pond to recover an object one has dropped. In the laboratory, these movement corrections and subsequent adaptation can be evoked and studied systematically using the classic visuomotor rotation[1,2] (VR) or force field[3] (FF) perturbation paradigms. In the VR paradigm, the subject receives distorted visual feedback−typically a rotation about the centre of the workspace−of a reaching movement, which creates a perceived error due to the mismatch of the expected and observed hand trajectory. In the FF paradigm, ongoing reaching movements are perturbed by imposing a force field that pushes the reaching hand away from the target, typically in a velocity-dependent manner. Humans can correct

their ongoing movements even during the very first trial after perturbation onset[4], a process that is mediated by the primary motor cortex (M1) integrating multiple inputs arriving from various sensory and motor brain regions[5–18].

When repeatedly exposed to a predictable perturbation, animals progressively learn to use their perceived errors to anticipate its effect. For the case of the VR paradigm described above, this leads to a gradual reaiming until the reach starts out in the correct direction[2], thereby eliminating the need for further online corrections. This adaptation process requires some form of rapid learning along the sensorimotor pathways, likely guided by trial-by-trial error information[4,19,20]. Although the question of how and where in the brain this motor adaptation happens remains inconclusive[1,21–46], and it may

[1]Department of Bioengineering, Imperial College London, London, UK. [2]Département de neurosciences, Faculté de médecine, Université de Montréal, Montréal, QC, Canada. [3]Mila (Quebec Artificial Intelligence Institute), Montréal, QC, Canada. [4]Department of Neuroscience, Northwestern University, Chicago, IL, USA. [5]Department of Biomedical Engineering, Northwestern University, Evanston, IL, USA. [6]Department of Physical Medicine and Rehabilitation, Northwestern University, and Shirley Ryan Ability Lab, Chicago, IL, USA. [7]These authors jointly supervised this work: Claudia Clopath and Juan A. Gallego. ✉e-mail: c.clopath@imperial.ac.uk; jgallego@imperial.ac.uk

depend on the characteristics of the perturbation[3,29,47–51], the widely accepted view is that motor adaptation combines several processes, including error calculation, sensory-based movement correction, and the update of an explicit forward "internal model" that predicts the sensory consequences of the ongoing action[22,23,50,52,53]. According to this view, motor adaptation is achieved by updating this forward internal model and then inverting it to define updated motor commands that successfully guide movement in the presence of the perturbation[52,54–57].

Recent work, however, has challenged this view: adaptation to a mirror reversal perturbation is more consistent with updating a "control policy" that prescribes the motor commands needed to attain a goal[57]. The authors further posit that this form of learning likely extends to many other perturbations. Thus, this new view eliminates the need to invert an updated forward internal model to achieve successful motor adaptation[54], proposing that learning is achieved by directly updating an existing control policy that maps intent into motor commands.

Here, we hypothesised that a recurrent circuit that controls its output based on an error-based feedback signal can leverage this same signal to adapt to a predictable perturbation through targeted synaptic changes (Supplementary Fig. 1). That is, we developed a model to test the feasibility of learning through control policy update without inverting an updated forward model. We chose the simplest model that would allow us to test this prediction: a single recurrent neural network (RNN) architecture. Since we were interested in learning from an error signal (as a means to achieve direct control policy update), we trained our single module RNN not to produce a pre-determined motor output, as virtually all studies in the field[58–67], but to control its output online using delayed feedback. This allowed us to address the hypothesis that feedback signals used for motor control could guide, in and of themselves, plastic changes within the network that would lead to successful trial-by-trial learning. We chose the online error as the teacher signal to guide learning, given recent evidence from human behaviour that fast feedback responses can potentially drive trial-by-trial adaptation to a persistent perturbation[68,69]. This error-based feedback signal was combined with the ongoing activity as part of a novel biologically plausible[70] plasticity rule that updated the recurrent weights trial-by-trial, seeking to minimise predictable motor errors.

We first show that an RNN can be trained to perform online motor control in the presence of a biologically plausible feedback delay[13]. We then demonstrate how delayed online feedback about the output can guide synaptic plasticity and successful trial-by-trial adaptation to a VR perturbation. Comparison with recordings of populations of M1 neurons collected as monkeys performed the same adaptation task (data from ref. [51]) supports the plausibility of the proposed plasticity rule, since the activity changes underlying adaptation in our model were also present in the actual neural activity. Analysis of the RNN activity changes during adaptation indicates that the different processes mediating trial-by-trial learning are intermingled at the single unit level, indicating that functionally distinct processes need not be implemented by equally distinct modules. Additional simulations provide causal evidence that our model did learn to counteract the effects of the perturbation by updating the control policy[57] that it acquired during initial training. Finally, our model adapted in a way that was similar to that of humans in terms of its time course[71,72], generalisation[2], and sensitivity to the variability of the perturbation[73,74], and replicated observations in monkeys that targeted manipulation of preparatory activity disrupts adaptation[44], suggesting that it captures several key aspects of learning. Thus, our work establishes that recurrent circuits that perform feedback-based error corrections can achieve trial-by-trial learning by directly updating their control policy, without the need for independent functional modules or inverting an explicit forward internal model.

## Results

### A recurrent neural network that performs feedback-based motor control

We built an RNN model to investigate whether the same feedback signals used to control ongoing behaviour could also enable motor adaptation. This work was divided into two phases: (1) training the RNN to perform feedback-based motor control (using a gradient-based algorithm) and (2) using this trained RNN to implement trial-by-trial motor adaptation via a local, biologically plausible plasticity rule acting on the recurrent weights of the network. We validated our model by comparing it to neural population recordings from monkey M1 during the same VR adaptation task we simulated[51].

First, we trained an RNN to produce desired movements (Fig. 1A, B). Our goal was to have a model that, after training, could dynamically adjust the ongoing movement based on incoming sensory feedback of instantaneous position error, $\epsilon_t$, defined as the difference between the current position, $p_t$, and the desired position, $p_t^*$, which we computed assuming a straight line between the start and end points (Fig. 1C). This error signal was fed back as an input to the RNN with a delay of 120 ms, based on estimates of visual system lags[13] (Fig. 1B).

After the initial training phase (Fig. 1D, examples in Supplementary Fig. 2; Methods), we tested the RNN on a standard centre-out reaching task with eight equally distributed targets. As expected due to our training procedure, the model was readily able to produce the required straight movement trajectories even without explicit training on this task (Fig. 1E).

To test the network's ability to correct its output, we replicated the classic VR paradigm[2] by rotating the visual feedback about the position of the "hand" by 30°. If the RNN could indeed control its output, it should still be able to reach the desired target by correcting the initial movement to counteract the 30° rotations. Inspecting the movement trajectories after VR onset confirmed that the model used the error signal to correct its ongoing output (Fig. 1F; the curved trajectories indicate ongoing correction). Importantly, successful correction relied on the error signal being fed back into the model: trajectory correction only started after the delayed feedback had had time to propagate to the model (note the discontinuity in the Fig. 1F paths), and an RNN trained without feedback connections (Methods) could not reach to the targets (Fig. 1G). Recurrent connections were necessary to handle time-delayed feedback (Supplementary Fig. 3), but when they were available, successful motor control could be achieved for a broad range of feedback (Supplementary Fig. 4) and recurrent (Supplementary Fig. 5) connection probabilities, in the presence of noise in any stage of the system (Supplementary Fig. 6), and for different control strategies (Supplementary Fig. 7). Thus, our model achieved robust online control beyond a specific set of architecture and hyperparameter choices.

### The same error signal used for feedback-based motor control can drive trial-by-trial adaptation

We have shown that an RNN that has learnt to use feedback signals to control its output can readily counteract an external perturbation, exhibiting a behaviour during the first trials following VR onset that is very similar to that of humans[2] and monkeys (compare the monkey data from ref. [51] in Fig. 2A to the model data in Fig. 2D). However, when repeatedly exposed to a VR, both humans and monkeys learn to adjust their initial "motor plan," which results in their reach take-off angle progressively moving toward the correct direction over tens of trials[2,51] (compare Fig. 2A, B). Can a recurrent circuit that performs feedback-based motor control use error signals to achieve similar trial-by-trial learning?

Since the feedback inputs acting on the network correctly modulate each unit's activity to minimise the ongoing motor error (Fig. 1F), we hypothesised that they could also act as a teacher signal for local, recurrent synaptic plasticity. To test this, we devised a biologically plausible local synaptic plasticity rule causing the connection weight

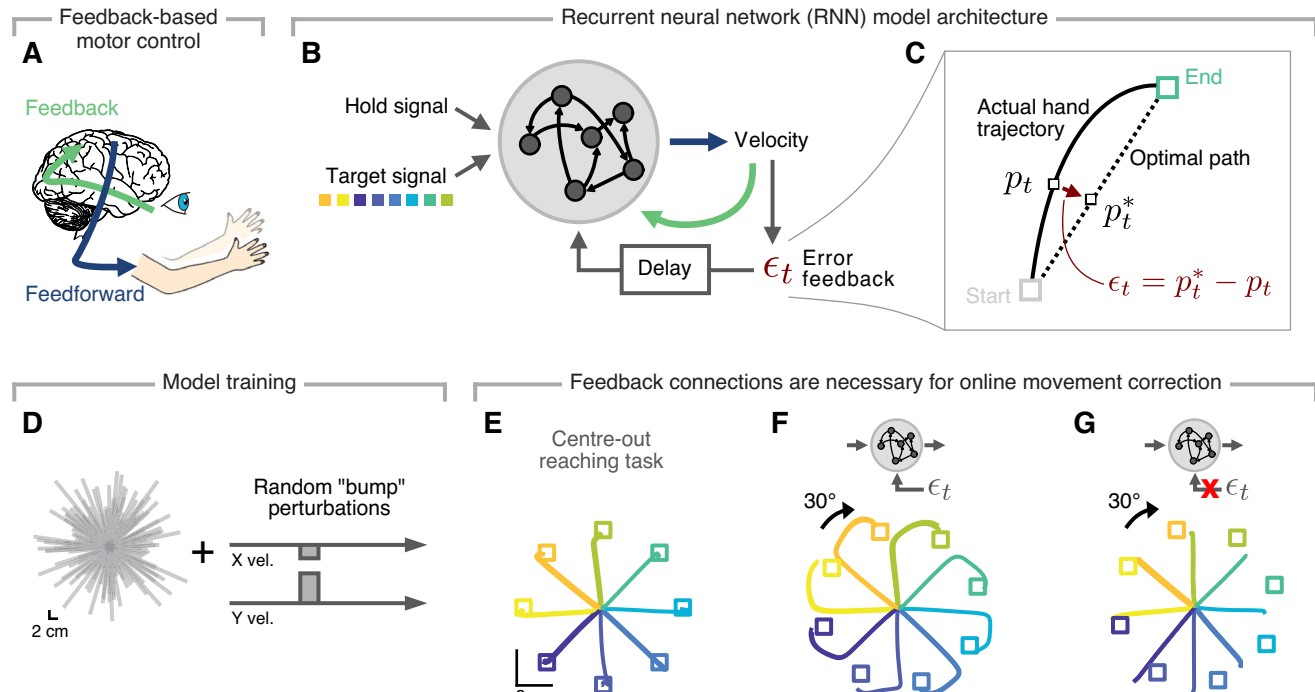

**Fig. 1 | Proposed recurrent neural network model that controls its output based on feedback. A** Feedback and feedforward pathways act together to produce precise movements, and to correct for perturbations. **B** A recurrent neural network (RNN) model to explore this shared implementation of online motor control and adaptation, each driven by a common error-based feedback signal. **C** We defined the ongoing error during movement as the difference between the observed and optimal hand position. **D** Initial RNN training included reaches of varying lengths and to different locations (grey lines) with occasional random velocity bump perturbations (cf. Supplementary Fig. 2). **E** Hand trajectories produced by an RNN that was trained to perform a standard eight-target centre-out reaching task. **F** Hand trajectories after introducing a 30° rotation of the RNN's output, to mimic a visuomotor rotation perturbation; note that feedback allowed the network to correct its output online, thereby reaching the target. **G** Hand trajectories for a model without a feedback loop, which could not counteract the same perturbation. The brain image is from NIAID Visual & Medical Arts. 06/27/2024. Lateral Brain. NIAID NIH BIOART Source. https://bioart.niaid.nih.gov/bioart/60.

from neuron $i$ to neuron $j$ to change in proportion to the feedback signal received by neuron $j$ (Methods). Implementing this plasticity rule led to behaviour like that of the monkeys: the initially large errors in take-off angle became progressively smaller over time, until they reached a plateau close to zero (note the similarities between Fig. 2D–F and Fig. 2A–C). Moreover, when the perturbation was turned off, the model underwent a de-adaptation phase similar to the "wash-out" effect exhibited by monkeys (compare the third and last epoch in Fig. 2C, F) and humans[75]. We observed the same behaviour across a very broad range of network connectivity parameters (Supplementary Fig. 4 and 5), when using either higher-dimensional (both position and velocity[76] rather than only position, Supplementary Fig. 7A–C) or less informative (endpoint, rather than the entire trajectory, Supplementary Fig. 7D–F) error signals, when adding noise at different stages (Supplementary Fig. 6A–I), and even when connectivity changes were made less localised (Supplementary Fig. 6J–L). These results confirm our main hypothesis: an error-based feedback process used for online motor control can, in and of itself, guide recurrent synaptic changes that drive successful trial-by-trial motor adaptation without the need to invert an updated explicit feedforward model, which our network lacked.

**The temporally dissociable activity changes that drive motor adaptation in the model can be uncovered in monkey primary motor cortex**

Our simulations suggest that feedback signals alone can be sufficient to drive rapid motor learning in a recurrent circuit. Can a similar type of learning be implemented in the neural circuitry of monkeys? Since it remains challenging to measure synaptic weight changes in vivo, we investigated the biological plausibility of the proposed form of learning

by characterising the activity changes in the RNN and comparing them to those in M1 neurons (data from ref. 51). We focused our comparison on M1 because it is a brain region strongly involved in both feedback-based movement corrections[8,14,17,38,77] and motor adaptation[43,51,78].

We devised an analysis that was based on the prediction that feedback-driven adaptation would be mediated by two processes, each of which would dominate at a different latency within a trial, and a different phase of learning (Methods). One process would dominate early in adaptation leading to activity changes due to feedback-based motor corrections late in each trial. The second process would be more prominent late in adaptation and mediate an updated reach direction (early in the trial), as a result of recurrent changes.

To measure activity changes due to these two processes, we computed the absolute difference between the single unit activities in two behavioural epochs after matching by target (Methods). First, to measure feedback-related changes (Fig. 3A; green), we probed the network in a state frozen right after perturbation onset in which we prevented adaptation-related weight changes (box B in Fig. 3A). We then compared this frozen-state activity to that in the baseline epoch (box A in Fig. 3A). Similarly, we measured learning-related changes (Fig. 3A; blue) by calculating the activity changes between the baseline epoch and the network frozen after 300 adaptation trials (box C in Fig. 3A). In this case, we expected the feedback component to have reached baseline levels again, as online movement correction would no longer be necessary (Fig. 2B, E). Finally, we defined the overall adaptation-related activity changes (Fig. 3A; dark grey) as the difference in activity between the early (box B in Fig. 3A) and late phases of adaptation (box C in Fig. 3A).

The overall RNN activity changes during adaptation (grey trace in Fig. 3C) included two distinct peaks within the trial, which we assumed

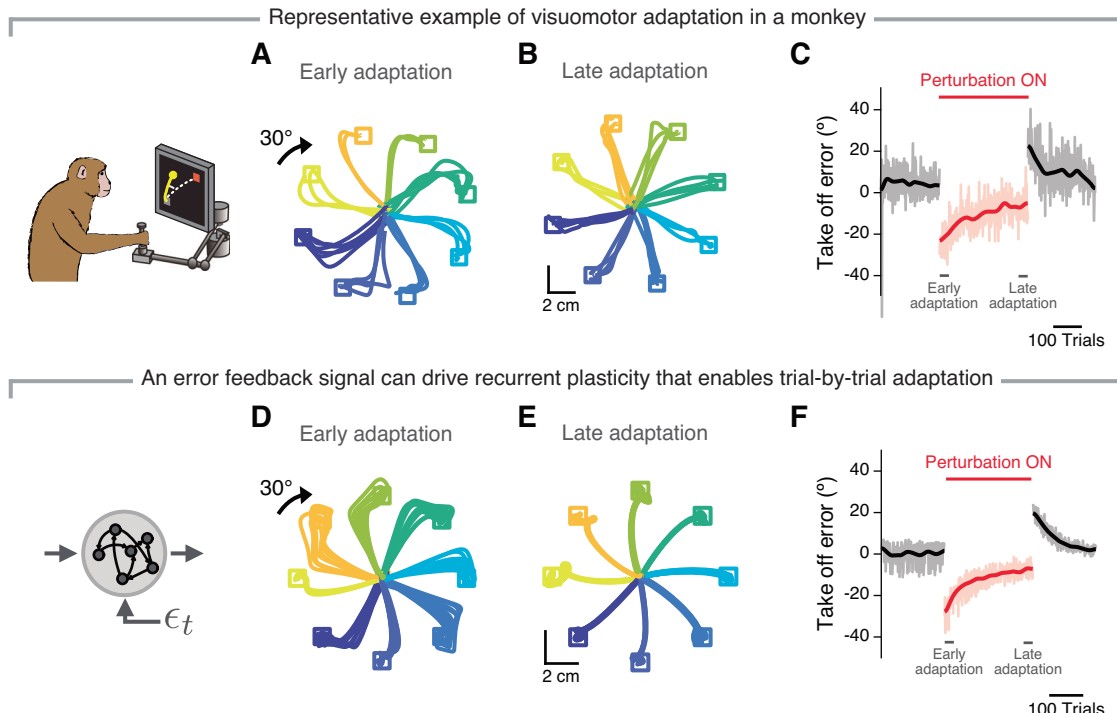

**Fig. 2 | Feedback signals can guide local synaptic plasticity that enables successful motor adaptation. A** Example hand trajectories as a monkey reached to each of eight targets after a 30° rotation of the visual feedback was introduced (first 30 trials after perturbation onset; data from ref. 51). **B** Example hand trajectories after the monkey had adapted to the perturbation by reaiming the reach direction (last 30 trials of the perturbation phase for the same session as in **A**). **C** Take-off angle error for the baseline (left, black), perturbation (middle, red), and wash-out phases (right, black). Transparent lines, single trial errors; solid lines, smoothed mean error (Gaussian filter, s.d., 10 trials). **D–F** Simulation results for an RNN implementing the proposed feedback-driven plasticity rule, by which recurrent weights are modified according to the error signal received by the postsynaptic neuron. The first and last 80 trials of adaptation are shown; otherwise data is presented as in **A–C**. Note the strong similarities between the behaviour of the network and that of the monkey. The monkey image was created by Carolina Massumoto, who gave permission to use it under a CC-BY license.

reflected learning (blue star, early during the trial) and feedback processes (green circle, late during the trial), respectively. To confirm this, we trained a different set of models that used gradient descent instead of feedback-driven plasticity to achieve motor adaptation (Methods). As expected, models trained using gradient descent also exhibited activity changes early in the trial (blue traces in Fig. 3C and Supplementary Fig. 8A), confirming that this first peak does represent a learning process. However, there was no second peak in these models, (Supplementary Fig. 8A), confirming that the later peak in the plasticity-driven models reflects feedback.

Having uncovered a signature of feedback-driven adaptation in our model, we sought to identify a similar change in neural population recordings from monkey M1 (Fig. 3B; data from ref. 51; Methods). Fig. 3D shows that the average change in M1 activity during a representative VR adaptation session included two temporally distinct peaks similar to those of our RNN model. To quantitatively compare the monkey data to our model, we calculated the ratio of the activity change at the times of the "feedback peak" (green circle in Fig. 3C, D) and the learning-related "feedforward peak" (blue star in Fig. 3C, D). For both monkeys, there was a substantial change in neural activity at the feedback peak during adaptation (Fig. 3F), similar in magnitude to the change in neural activity at the learning peak, as predicted by our model (Fig. 3E). Interestingly, when we repeated our analysis for dorsal premotor cortex (PMd; data recorded simultaneously with M1, also from ref. 51), we observed a much weaker feedback-related activity peak than in M1 (Supplementary Fig. 8D, F), which is consistent with PMd being less driven by somatosensory feedback than M1, perhaps due to its higher position in the neuraxis[14]. Thus, although our model was not designed specifically to represent M1, it may nonetheless be most functionally similar to M1, since this region is strongly involved in most

aspects of sensorimotor control[8,38,77]. Finally, we compared activity changes following adaptation across models with different feedback projection densities. In agreement with experimental observations reporting that as many as 73% of M1 neurons exhibit feedback responses[17], models with larger feedback projection density (~80–100%) had activity changes that were consistently closest in magnitude to those observed in the monkeys (Supplementary Fig. 8B, C). Overall, the similarities in the activity changes during adaptation between our model and monkey M1 show that error-based feedback signals may indeed guide local plasticity that enables rapid motor learning in behaving animals.

## Recurrent neural networks learn by updating their control policy through functionally distinct processes that are intermixed in their implementation

After having established that key features of the activity changes predicted from RNNs implementing feedback-based learning were recapitulated in monkey M1 (Fig. 3), we studied how RNN models achieved adaptation by taking advantage of our full access to their activity and connectivity. A trial-by-trial analysis of the activity changes during adaptation confirmed that feedback-driven corrections dominated the early stage of adaptation, whereas learning-related changes lead to a reaiming of the output[2,72] became apparent after ~20 trials of this initial feedback-driven phase (Fig. 4A–C). A substantial amount of units showed both feedback-related and learning-related activity changes (57% of the units for the example network in Fig. 4A, B; between 46 and 60% of the units across all 10 different network simulations), suggesting that these two distinct processes might be intermixed in their implementation by the RNN.

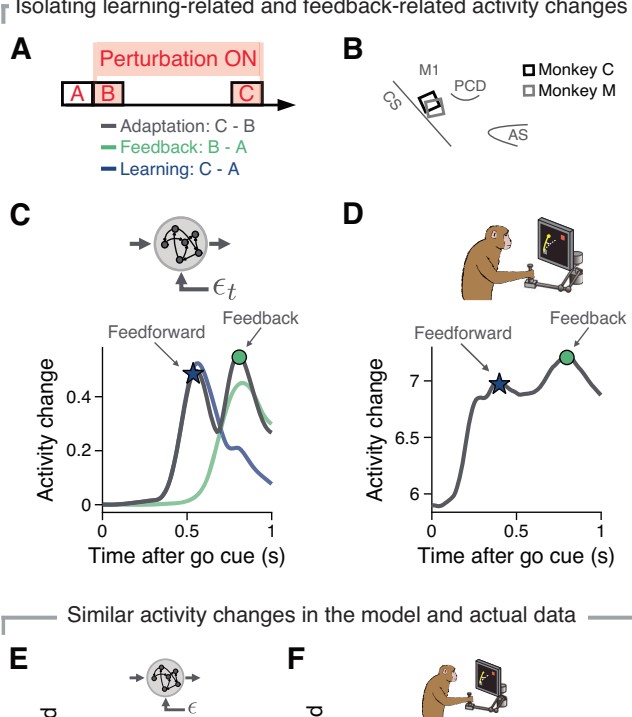

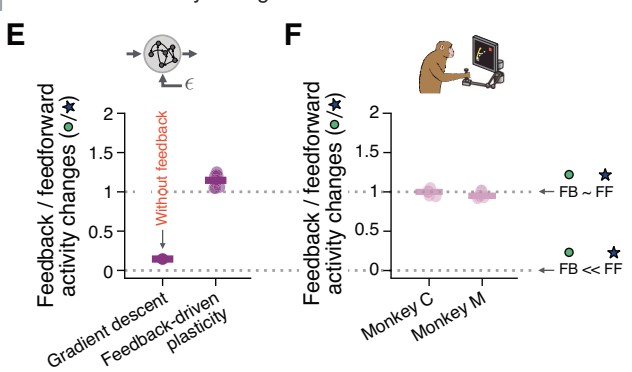

**Fig. 3 | Two temporally dissociated activity changes may indicate feedback-driven plasticity in monkey M1 during VR adaptation. A** Epochs used to identify the overall adaptation-related activity changes (grey), feedback-related activity changes (green), and learning-related activity changes (blue). **B** Approximate recording array location for each of the two monkeys (legend). **C** The average RNN activity change across pairs of behavioural epochs (as shown in **A**) reveals putative feedforward-related (blue star indicates the defined peak time) and feedback-related (green circle indicates the defined peak time) activity changes for an example network. Activity change is measured as an absolute difference. Black trace, activity change between perturbation onset and successful adaptation; green, activity change between baseline and perturbation onset; blue, activity change between baseline and successful adaptation. Note that the peak timings (star and circle) are chosen based on ten different network simulations, and hence do not overlap perfectly with the traces from the example simulation. **D** The average adaptation-related activity change of M1 neurons between perturbation onset and successful adaptation includes two temporally dissociable peaks, similar to the network data. Data from one representative session from Monkey C. **E** Ratio of feedback-related (green circle in **C**) to feedforward-related activity changes (blue star in **C**) in models trained without feedback using stochastic gradient descent (left), and models learning on a trial-by-trial basis using our feedback-driven plasticity rule (right). Individual markers, ten simulations; horizontal line, mean. **F** Ratio of feedback-related activity changes (green circle in **D**) to feedforward-related activity changes (blue star in **D**) for monkey M1 (data shown for each monkey separately, five sessions for monkey C, and six sessions for monkey M). Individual markers, individual experimental sessions; horizontal line, mean. FB feedback, FF feedforward. The monkey image was created by Carolina Massumoto, who gave permission to use it under a CC-BY license.

We verified this observation by establishing that, for this group of units, the magnitude of the feedback-driven activity changes exhibited by a unit early during adaptation was related to the amount of learning-related activity changes at the end of adaptation, as would be expected from our plasticity rule (Methods). Indeed, a correlation analysis between the average feedback-related activity change in the first 30 adaptation trials and the average learning-related activity change in the last 30 adaptation trials for each unit confirmed a significant association ($r = 0.57$, $P < 0.0001$, $n = 229$, Fig. 4D) between the implementation of those processes by single network units.

Interestingly, we found that even if most units (~80%) showed learning-related changes, only a subset of them contributed to feedback corrections (~60%), which seems to be learnt during the initial training phase (Supplementary Fig. 2J–L). By examining the evolution of the single unit activity during initial training, we posit that our networks acquired a subset of units that did not respond directly to the incoming feedback to stabilise their dynamics and prevent oscillations in the output due to the delayed error feedback, since these were only present during the very initial phases of learning (Supplementary Fig. 2). Combined, these analyses indicate that the feedback-driven and learning processes share a common implementation at the single unit level, which likely evolved during the initial training phase.

Finally, we tested whether our RNNs learnt by updating the control policy they implemented[57], as we anticipated based on their architecture, and plasticity rule. We first devised an analysis that captured the input-output mapping (i.e. the control policy) that a network acquired during initial training (Fig. 1D). Since during baseline trials, activity follows well-defined patterns, we expected to be able to predict future movement from ongoing activity. This is indeed what we found: simple linear "decoders" could predict future endpoint velocity well during baseline (Fig. 4E in black; note that predictions degraded in models without recurrent connections, Fig. 4E shown in orange, as did these networks' ability to handle delayed feedback, Supplementary Fig. 3A–F). Did this control policy get updated during adaptation, as expected from our analysis of trial-by-trial activity changes underlying learning (Fig. 4A–D)? To test this, we asked whether the performance of decoders trained to predict future output from the ongoing activity of fully adapted RNNs became progressively worse if they were tested earlier during adaptation[51]. As predicted, the performance of these decoders degraded as one went back in time to the beginning of adaptation (Fig. 4F), indicating that networks updated their mapping between ongoing activity and motor output during adaptation. This suggests that the control policy that RNNs acquired after initial training was updated during adaptation.

This analysis, however suffers from one potential confound: our networks, as robust controllers, may have acquired during initial training a forward model that predicted the consequences of their motor output[79], and updating such forward model during adaptation could account at least partly for the decrease in decoder performance during early adaptation shown in Fig. 4F. To establish more directly that our networks learnt by updating their control policy, we performed an additional set of simulations that provided causal proof for this prediction. In these simulations, networks had to adapt to a VR perturbation that matched the angle between two consecutive targets (45°). Consistent with the notion that our networks learnt based on direct policy update, when we provided fully adapted networks with a cue that during baseline led to a reach to a specific target, (e.g. to the 0° target), after adaptation, they would reach toward the target right next to it (e.g. the −45° target; see Fig. 4G). Thus, our networks learnt to counteract perturbations by updating their control policy through two

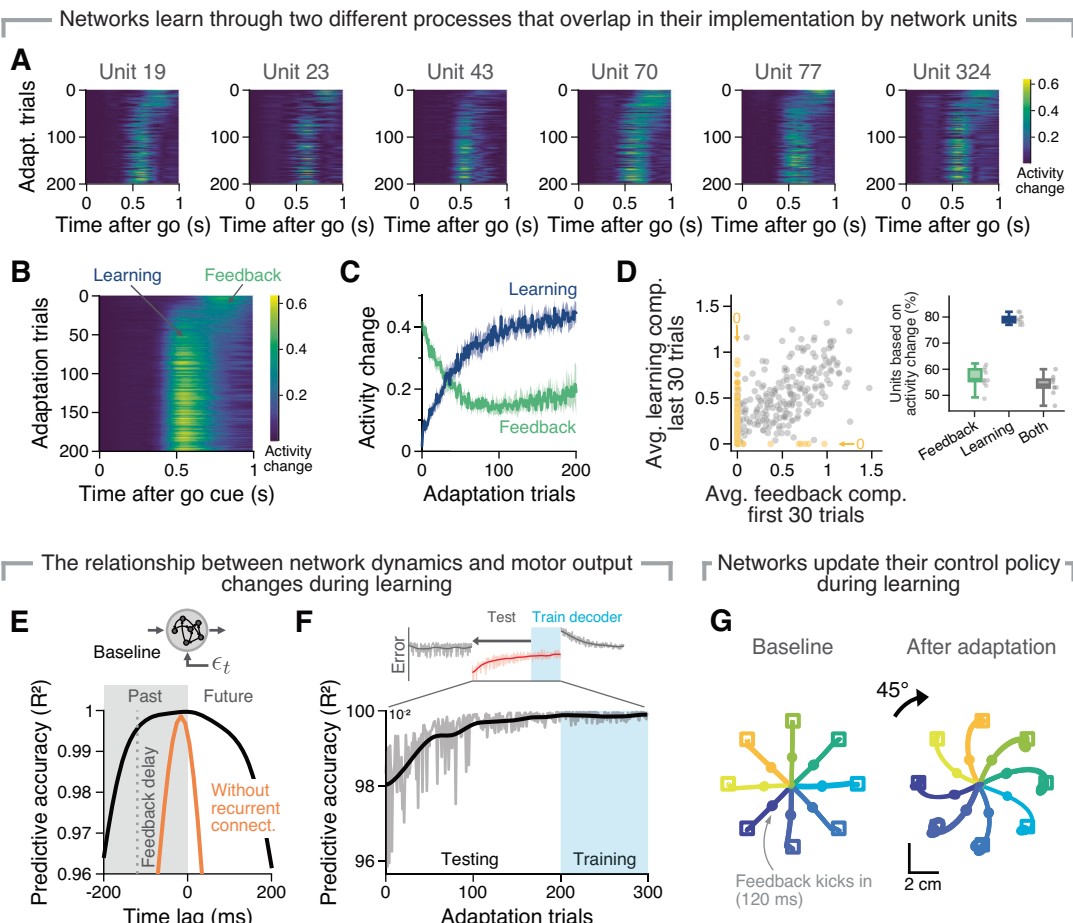

**Fig. 4 | Characterisation of adaptation in the network model. A** Activity change to respective baseline trials of example units during adaptation. Single units switch from showing feedback-related (activity change around 800 ms after go) to showing learning-related (activity change around 500 ms after go cue) activity changes. Activity change is measured as an absolute difference. **B** Average of all neurons in an example network. Data presented as in (**A**). **C** Time course of average activity change (**B**) for two fixed time windows within the trial at 500 ms (green) and 800 ms (blue) after go cue across all ten simulated networks. Line and shaded area, mean ± s.d. **D** Correlation between the feedback-related activity changes during the first 30 adaptation trials and the learning-related activity changes during the last 30 adaptation trials for all the units from one representative network that exhibited both types of activity changes ($r = 0.574$, $P < 0.00001$, Pearson's correlation, $n = 229$ units; a similar correlation was observed in all networks). Right inset: percentage of units that showed feedback-related activity changes (green), learning-related activity changes (blue), or both types of changes (dark grey) across all ten simulated

networks. Boxplots show median, centre line; interquartile range, box; and data range (minimum to maximum), whiskers. Grey markers, individual networks. **E** Accuracy of a decoder trained to predict time-lagged velocity output for two example networks with (black) and without (orange) recurrent connections (cf. Supplementary Fig. 3B). Recurrent connectivity is necessary for accurate prediction over a timescale that includes the feedback delay (dotted grey line). **F** The accuracy of a decoder trained on the last 100 trials of adaptation degrades when tested in earlier trials, which is consistent with an update of the learnt control policy. Grey lines, single trial accuracy; black trace, smoothed accuracy (Gaussian kernel; width, ten trials). **G** Example hand trajectories before and after adaptation to a 45° visuomotor rotation show that networks learn by updating their control policy: specific inputs (colour-coded targets) are mapped onto different motor outputs (colour-coded trajectories), leading to reaches that are mostly aimed to the target that was situated next to the original target. Coloured circular markers, a moment in which feedback input starts arriving at the network after the 120-ms delay.

distinct processes that were overlapping in their implementation by single units.

## Feedback-based learning recapitulates additional features of behavioural adaptation

The previous results suggest that a relatively simple plasticity rule based on an error feedback signal may mediate motor adaptation by a recurrent circuit, and that this form of learning may be recapitulated in monkey M1. A potential concern is that this type of learning may only apply to the particular perturbation experiment we have modelled so far. To address this, we investigated whether our model replicated a broader range of behavioural observations from human and monkey motor adaptation studies.

We first addressed the finding that humans learn more from a given trial if they experience a larger error[68]. Our model reproduced this trend: the measured correlations between movement error and

amount of learning in the next trial were comparable in magnitude and sign to those of monkeys performing the same VR task (Fig. 5A; data from ref. 51). Moreover, as in human adaptation studies[73,74], our model's ability to learn was also hindered when the perturbation was inconsistent across trials, with greater perturbation variance leading to progressively less learning (Fig. 5D). Thus, the amount of trial-by-trial adaptation matched experimental observations well.

The motor system adeptly generalises what it has learned to many novel situations; the amount of generalisation seems to depend on the similarity between the current and the past situation. During a VR experiment, participants who have adapted to a perturbation applied on only a single reach direction generalise when reaching to neighbouring targets, to an extent that decreases as the angle between the new and adapted direction increases[2]. Repeating this single-target adaptation experiment in our model revealed a similar generalisation pattern: the model readily

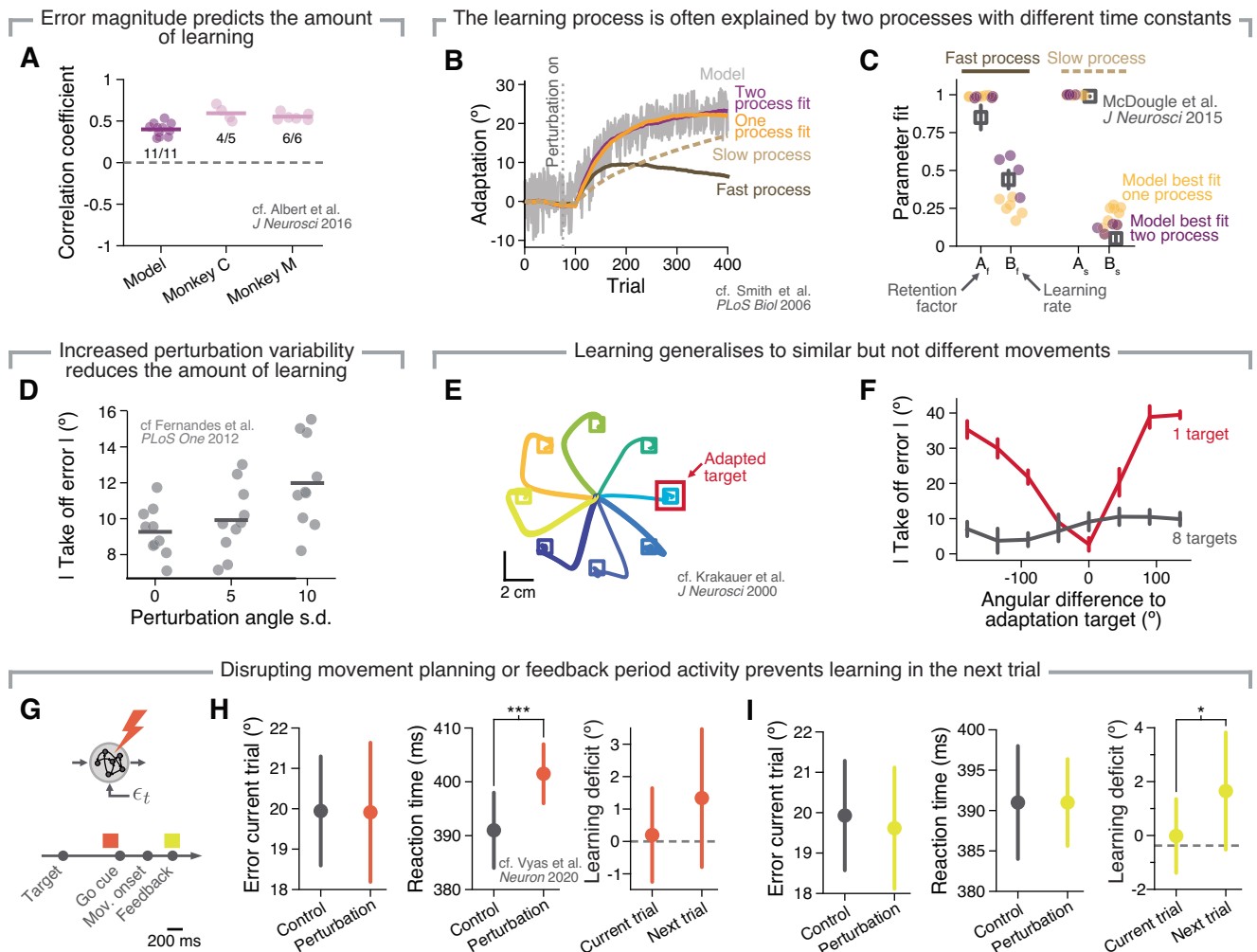

**Fig. 5 | Motor adaptation based on feedback-driven plasticity recapitulates key aspects of human and monkey behaviour. A** Correlation between the take-off error in the current trial and the amount of learning from this trial to the next for both simulation (Model) and behavioural data (Monkey C and Monkey M; data from ref. 51). Individual circles, a different network (Model) or experimental session (Monkey); numbers, the proportion of networks or experimental sessions exhibiting a significant correlation ($P < 0.05$). **B** As in human experiments, the time course of adaptation in the model is often best described by a dual-rate model[71]. Grey line, single trial errors of simulated adaptation behaviour in an example simulation. Purple line, fit for the dual-rate model; golden line, fit for single-rate model; dark and light brown lines, fast and slow processes, respectively, for the dual-rate model. **C** Fit parameters of the single (golden) and dual-rate model (purple) that best capture the behaviour of each model. Note that dual-rate parameters match those from a visuomotor adaptation study in humans[72] (dark grey).

Individual circles, ten different networks; square and error bars, mean and 95% confidence interval. All four parameters for dual-rate models are shown even when the best fit is provided by a single-rate model, to illustrate that quality of fit can be distinguished mostly based on the value of $B_f$ and $B_s$. **D** Movement error after adaptation to VR perturbations with different perturbation angle variability (cf. refs. 73,74 for human behaviour). **E** Hand trajectories produced by the model after visuomotor adaptation to a single target ("Adapted target", in red). **F**. Take-off error after adaptation to visuomotor perturbations applied to a single target, as shown in E (red), or to eight targets (dark grey) (cf. ref. 2 for human behaviour). Lines and error bars, mean and s.d. across ten networks. **G–I** Perturbations to network activity at different time windows within a trial hinder learning in the next trial (cf. ref. 44 for monkey behaviour). Perturbations were applied at one of two epochs: before the go cue (**H**), and around feedback response (**I**).

anticipated perturbations applied on adjacent targets, and adaptation decreased as the angle between the probed and adapted target increased (Fig. 5E, F).

When examining the timescale of learning during an experimental session, human motor adaptation seems to be mediated by two simultaneous learning processes: one fast, and another slow[71,72]. Using the same analysis as in ref. 71 (Fig. 5B), we found that the adaptation time course of our model was often best described by a combination of two learning processes with different time constants (purple markers in Fig. 5C), which generally took similar values to those reported for humans[72]. However, for a subset of models, a single process was sufficient (golden markers in Fig. 5C). We hypothesised that the difference in learning timescales across models that only differed in their randomly chosen initial connection weights may have been driven by

differences in their recurrent connectivity, since it was the interaction between the recurrent activity and output errors that drove trial-by-trial learning. Indeed, when we reduced the recurrent connection probability to 50%, the adaptation process was almost always best captured by models with a single timescale, much more frequently than for our standard models with 100% recurrent connectivity (Supplementary Fig. 9A–C). This confirmed the intuition that the more complex recurrent connectivity of networks with 100% connection probability led to a more complex adaptation behaviour. As expected, models showing two learning processes with different timescales did not exhibit spontaneous recovery during trials in which the error was "clamped" to zero (replicating the experimental protocol from ref. 71), suggesting that these two observations are not necessarily linked and should be carefully examined independently.

Finally, it has been recently shown that disrupting movement planning hinders learning in a very specific way[44]. The authors delivered subthreshold intracortical microstimulation to either PMd or M1 as monkeys planned the upcoming movement under a VR perturbation, and observed decreased learning in the next trial, even if the motor output in the current trial remained unaffected. We replicated this experiment by selectively disrupting our RNN's planning activity by injecting an extra input to all units in the network during the delay period, in a manner that was analogous to that of ref. 44 (Fig. 5G). In good agreement with the monkey experiments, this stimulation did not affect the motor output in the current trial, but significantly increased reaction time, and decreased the amount of learning in the next trial (Fig. 5H). In addition, our simulations predict that delivering a perturbation during the feedback time window will also lead to a learning deficit in the next trial (Fig. 5I; additional predictions in Supplementary Fig. 9D–G). Combined, these five additional "experiments" indicate that our model reproduced many key features of human and monkey adaptation, supporting the view that a purely feedback-driven policy update process may mediate trial-by-trial adaptation.

## Discussion

Both the rapid correction of ongoing movements and progressive adaptation to changing conditions are key landmarks of behaviour that are often studied separately. Here, we have shown that an RNN that can dynamically control its output based on error-based feedback inputs can use those same inputs to achieve trial-by-trial motor adaptation through recurrent connectivity changes alone. Further, adaptation is driven by two distinct processes—feedback-based corrections and progressive learning—that are jointly implemented by the same units. This form of feedback-driven plasticity led to identifiable activity changes in the RNN that recapitulated key features of neural recordings made in monkey M1[51], and human[2,44,71–74] and monkey adaptation behaviour[44]. Our results thus extend recent experimental studies[57,68,80] and suggest that a variety of phenomena characterising trial-by-trial motor adaptation may result from the internal properties of error-driven neural circuits that update their control policy to minimise ongoing errors.

Typically, modelling studies on motor control have focused on understanding it at an abstract computational level, largely ignoring neurons and connections between them[68,71,81]. One potential reason is the challenge of mapping the abstract concepts of optimal feedback control theory into brain regions[6,18]. Indeed, even if the brain clearly has a certain degree of modularity as it is most evident when examining the shared signs exhibited by different individuals who have suffered an injury in the same brain region, experimental evidence challenges traditional approaches that seeks to map distinct computations into distinct brain regions[80,82,83]. Our work differs from those previous attempts in that it approaches the problem from a bottom-up, not a top-down perspective: instead of explicitly training neural networks to implement the behavioural processes predicted by optimal feedback control theory, such as a forward model, a state estimator, or a controller[84], we let error minimisation guide the emergence of an efficient control strategy, potentially mimicking how brain connectivity developed over evolutionary timescales[85–87]. This bottom-up approach led to adaptation that recapitulated key aspects of human and monkey studies (Figs. 2, 5), providing support to the view that distinct computations need not map onto distinct circuits or regions.

Our work also provides insight into the processes that are necessary for successful motor adaptation. A recent study[57] proposed that motor adaptation is not driven by inverting an updated forward model that predicts the sensory consequences of actions to issue adapted motor commands[54]. Instead, the authors argue, motor adaptation is driven by the direct update of a control policy. This is precisely how our model achieved trial-by-trial adaptation: by adapting the control policy it learned during initial training (Fig. 4). It could not learn by inverting an updated explicit forward model, because it lacked one. Interestingly, a decoder analysis suggests that during initial training, our networks did learn in their dynamics a forward model that predicted the consequences of their ongoing action (Fig. 4E, F), but this is to be expected from a robust controller[79], especially if the feedback is delayed.

In addition to validating our model based on its ability to replicate behavioural results (Fig. 5), we further tested it by comparing its activity to neural recordings from monkeys performing the same VR adaptation task[51]. When compared to M1 (Fig. 3) and PMd (Supplementary Fig. 8) activity, our models most closely resembled monkey M1. This likely happens because while both regions show clear changes during learning, M1 is more involved in feedback-based corrections than PMd[14,77]. However, this does not necessarily mean that our model encompasses the M1 function alone. On the contrary, it is likely that it captures functions mediated in part through multi-region interactions with a variety of cortical and subcortical regions[26,88–91], and spinal circuits[80], such as feedback processing, or sensory gating[92]. Extending our work to modular, multi-region RNNs (based on previous studies, including refs. 63,65,66,93) that control a more realistic plant that also includes the various afferent pathways that are key for movement generation[12,80] may shed light into the distributed implementation across neural circuitry of the different processes underlying feedback control and motor adaptation.

The cerebellum is of particular interest, given its long-held key role in motor adaptation[6,23,26,32,42,94–99]. The central premise of these earlier studies was that the cerebellum stores both forward and inverse internal models of the sensorimotor system[50,52,55,56], and that adaptation is based on inversion of an updated forward model[54]. The adaptation deficits in cerebellar patients[100] were taken as important support for this hypothesis. As mentioned above, this internal model-based view of adaptation has recently been challenged by a proposal that adaptation may be better described by a direct update of a control policy[57], and our work provides evidence of how neural circuits could implement this strategy. Yet, the cerebellum could readily fit into our model as the region responsible for calculating error estimates based on the sensory feedback, which we simply assumed to exist, and/or as a key node in the implementation and/or update[26] of the control policy. Interestingly, the fact that cerebellar patients have impairments in motor control as well as learning[26,101] also fits with our model, since both processes are critically dependent on the availability of an accurate error signal and an appropriate control policy. The fact that adaptation became slower when noise was added to the error signal (compare Supplementary Fig. 6G–I to Fig. 2D–F) illustrates the impact that inaccurate feedback has in our model. Extending our bottom-up model of feedback-based learning so it also calculates the ongoing error would allow us to test this prediction, and continue investigating the hypothesis that motor adaptation is based on a policy update process[57].

Learning in our model is based on a new feedback-driven plasticity rule, which adds to recent efforts to implement biologically plausible learning in RNNs[70,102–107]. The key challenge here is solving the so-called temporal credit assignment problem: since any change in recurrent connectivity not only affects the output at a given time step but can also influence the whole time course of the ongoing dynamics, it is difficult to predict how to modify a recurrent weight in order to shape the output in a desired way. Most of the existing biologically plausible plasticity rules share the same basic idea: the weight change is proportional to the error signal arriving at the postsynaptic neuron times the activity history of the presynaptic neuron. What makes those plasticity rules biologically plausible is that both pieces of information could, in principle, be locally available at a synapse[70]. For example, in the cerebellum, climbing fibres provide inputs that can lead to learning-related changes in Purkinje cell activity, whereas mossy fibres provide a separate source of sensory information. Functionally dissociable inputs may also be available to cortical neurons, since local axonal projections tend to arrive at proximal dendritic regions, whereas top-down inputs impinge on distal

dendritic regions[108]. The latter can produce nonlinear dendritic events, which strongly modulate plasticity[109]. Finally, an alternative−or perhaps additional−way in which these functional dissociable inputs could be available to cortical regions is via layer-specific projections, as proposed in studies focusing on the implementation of predictive coding theories[110]. This way, our model could be functionally related to how predictive coding computations are implemented by the brain.

The crucial difference between our model and previous investigations of biologically plausible plasticity rules is that the error signal is a direct input to the neuron, as opposed to only a signal used for learning. This allows it to simultaneously guide weight updates and affect the ongoing network dynamics. This feature is desirable because it avoids the need to have two distinct pathways for error signals and ongoing network dynamics, respectively[111–113]. Moreover, our approach makes weight update dependent on the error signal in two ways: directly, through the error term in the plasticity rule, and indirectly, through the change in ongoing dynamics, which influences the activity of the pre-synaptic neuron. This could be beneficial for learning since our plasticity rule approximates the gradient on a trial-by-trial basis, as we have shown in a recent theoretical study[114].

To conclude, we have shown that a recurrent circuit can use the same error-based feedback signal to both correct its motor output and achieve trial-by-trial motor adaptation through guided synaptic plasticity. The feedback-related and learning-related processes underlying adaptation were jointly implemented by the same units, without the need for distinct functional modules. Moreover, all features of our model emerged naturally after initial training and did not need to be engineered top-down. Despite this, our model recapitulated key observations from behavioural and neurophysiological adaptation studies in humans and monkeys. Adaptation was achieved by directly updating the control policy that mapped network inputs into motor output, supporting the view that motor learning can be achieved without inverting an updated forward model, which our model lacked. Thus, recurrent circuits can leverage the same error feedback signal to adjust behaviour on multiple timescales, both through online movement correction and trial-by-trial adaptation.

## Methods
### Recurrent neural network model

Neural activity $y$ was simulated using the following dynamical equations,

$$x_j(t+1) = x_j(t) + \frac{dt}{\tau}\left(-x_j(t) + \sum_i W_{ji}y_i(t) + \sum_i W_{ji}^{in}s_i(t) \right.$$
$$\left. + \sum_k F_{jk}\epsilon_k(t-\Delta) + b_j\right) \quad (1)$$

$$y_i(t) = \Phi(x_i(t)) \quad (2)$$

$$r_i(t) = \sum_{t'<t}^{T} y_i(t') \quad (3)$$

$$v_k(t) = \sum_i W_{ki}^{out}y_i(t) + b_k^{out} \quad (4)$$

$$p_k(t) = p_k(t-1) + dt \cdot v_k(t) \quad (5)$$

$$\epsilon_k(t) = p_k^*(t) - p_k(t) \quad (6)$$

where the network output $v$ represents velocity and $p$ position of the simulated planar hand movement (cf. definitions in Table 1). The instantaneous error signal $\epsilon$ is given by the difference between the target $p^*$ and the produced position $p$, and is fed back to the network with a time delay $\Delta$ (except in the control simulation without feedback in 1G). The way we constructed the network input $s$ and the target position $p^*$ is described in the 'Reaching datasets for model training and testing' section below. Each trial was initialised by setting all $x_j$ to

**Table 1 | Simulation parameters**

| Parameter | Definition | Value |
|---|---|---|
| dt | Time step | 10 ms |
| $\tau$ | Time constant | 50 ms |
| $\Delta$ | Feedback delay | 120 ms |
| N | Number of neurons | 400 |
| $\Phi$ | Nonlinearity | ReLU |
| y | Neural activity | – |
| s | Stimulus | – |
| $\epsilon$ | Position error | – |
| r | Eligibility trace | – |
| W | Recurrent weight matrix | – |
| b | Recurrent offset | - |
| $W^{in}$ | Input weight matrix | – |
| $W^{out}$ | Output weight matrix | – |
| $b^{out}$ | Output offset | – |
| F | Feedback weight matrix | – |
| v | Velocity (2D) | – |
| p | Position (2D) | – |
| $\alpha$ | Gradient Descent: Learning rate | 0.001 |
| B | Gradient Descent: batch size | 20 |
| $\beta$ | Gradient Descent: weight regularisation | 0.001 |
| $\gamma$ | Gradient Descent: activity regularisation | 0.002 |
| $\eta$ | Feedback-driven plasticity: learning rate | 0.00002 |
| T | Reach trajectories: number of time steps in a trial | 300 |

random numbers uniformly distributed between −0.2 and 0.2. All simulations were performed on RNN models consisting of 400 units, which were connected all-to-all. Varying the recurrent connection probability did not change the results (Supplementary Fig. 5). Subindexes $j$ and $i$ indicate units in the network, whereas $k$ indexes the output dimension ($x, y$).

**Initial model training procedure.** The first step was to train the RNN to control its own output, that is, to minimise the position error, $\epsilon$. This initial training was performed using standard gradient descent to find the right set of parameters. We implemented it in Pytorch[115], using the Adam optimiser with learning rate $\alpha = 0.001$ ($\beta_1 = 0.9$, $\beta_2 = 0.999$)[116]. The weights ($W, W^{in}, W^{out}, F$), and biases ($b, b^{out}$) were initialised by drawing random, uniformly distributed numbers between $-1/\sqrt{l}$ and $1\sqrt{l}$, where $l$ is either the number of units in the network (for $W, W^{in}, F, b$), or the dimensionality of the output (for $W^{out}, b^{out}$). The gradient norm was clipped at 0.2 prior to the optimisation step. The loss function used for this initial training phase was defined as

$$L = \frac{1}{2BT}\sum_b^B\sum_t^T\sum_{k=x,y}\epsilon_k^2(t,b) + \beta\sum_{M=(W,W^{in},W^{out},F,b,b^{out})}||M||_2 + \gamma\frac{1}{NBT}\sum_b^B\sum_t^T\sum_i^N y_i(t,b)^2 \quad (7)$$

where $B$ is the batch size, $T$ the number of time steps, $N$ the number of units, $\beta$ the regularisation parameter for the weights and bias terms, and $\gamma$ the regularisation parameter for the activity in the network (cf. definitions in Table 1). The network was trained for 1100 epochs, divided into three blocks of different lengths (100, 500, 500; examples in Supplementary Fig. 2, where they are referred to as "phases"). For the first 100 epochs, the feedback weights $F$ were kept fixed while the remaining parameters were allowed to change. This ensured that the model learnt to self-generate the appropriate dynamics to produce a variety of reaching trajectories. In the next 500 epochs, the feedback connections were also allowed to change. In the last 500 epochs, we

introduced perturbations on the produced output (see "Reaching datasets for model training and testing"), while keeping all parameters plastic, to make the model learn to use the feedback inputs to compensate for ongoing errors. We initially tested different training schedules and discovered that it is critical to have an initial phase where the model learns to perform feedforward motor control (the first 100 epochs with fixed feedback weights), and a later phase where it acquires the ability to react to perturbations using feedback control (the last 500 epochs with perturbations). The number of epochs for each phase was chosen after qualitative assessment of the produced output trajectories.

**A feedback-driven plasticity rule to drive trial-by-trial learning.** Having set up the model to control its own output, we next examined how error-based feedback inputs $\epsilon$ could guide learning, implemented through synaptic plasticity within the recurrent weights $W$ of the network. To this end, we devised the following feedback-driven plasticity rule:

$$d\tilde{W}_{ji}(t) = \mathrm{dt}\, \eta \sum_{k=x,y} F_{jk}\epsilon_k(t-\Delta)r_i(t) \quad (8)$$

$$W_{ji}(\text{next trial}) = W_{ji}(\text{current trial}) + \sum_{t}^{T/5} d\tilde{W}_{ji}(5t) \quad (9)$$

The weight update $d\tilde{W}$ was calculated online and summed up taking into account every fifth time step until the end of a trial[64,117]. We did this to illustrate that learning can happen on a more coarse-grained timescale than the original neural dynamics. After each trial, we applied this accumulated weight change and updated the recurrent weights $W$ accordingly. Our plasticity rule for feedback-based learning was inspired by other studies[70,102–107] that have been shown to approximate gradients of the mean squared error loss $L_{\mathrm{MSE}} = \frac{1}{2T}\sum_t^T \sum_{k=x,y}\epsilon_k^2(t)$ in other models.

Note that the proposed plasticity rule differs from standard gradient descent in three key aspects:

1. We use learnt feedback weights $F$, instead of $\left(W^{\mathrm{out}}\right)^T$ [118].
2. The error signal $\epsilon$ is delayed by $\Delta$ to simulate delayed feedback in biological circuits, and it is a real input to the units in the network.
3. There is no error backpropagation through time; instead, we use an eligibility trace $r$ (an accumulation of ongoing activity; eq. (3))[70].

**Reaching datasets for model training and testing.** The network model was trained to produce a broad set of synthetic planar reaching trajectories following an instructed delay phase. The $x$ and $y$ positions of the starting ($p^{\mathrm{start}}$) and ending points ($p^{\mathrm{end}}$) of those trajectories were randomly drawn from a uniform distribution ranging from −6 cm to 6 cm. To simulate natural reaching behaviour, we interpolated between these points using a sigmoid function

$$f(t) = \frac{1}{1 + \exp(-t \cdot \kappa)} \quad (10)$$

where $\kappa = 10$ cm/s. The manually constructed reach trajectories were thus given by

$$p_t^* = p^{\mathrm{start}} + f(t)(p^{\mathrm{end}} - p^{\mathrm{start}}) \quad (11)$$

which resulted in bell-shaped velocity profiles.

Each trial lasted 3 s and included an instructed delay period, randomly drawn from 0 to 1.5 s.

The network received an input signal consisting of a two-dimensional target signal and a one-dimensional timing signal. The target signal, defined as ($p^{\mathrm{end}} - p^{\mathrm{start}}$), was delivered to the network 0.2 s after trial onset, and kept fixed until the end of the trial. The timing signal (referred to as "Hold") was given in the form of a constant, and was switched to zero at the time corresponding to the "go" signal, which varied between 0.2 and 1.7 s for the random reaching task used during training (that is, when the network generated reaches of

random direction and lengths up to 8.5 cm), and between 1.2 and 1.7 s for the centre-out-reaching task.

As mentioned above, during the last phase of the initial training phase, we included brief bump perturbations to the output of the network, so it had to learn to use the feedback input to correct its output online. In 75% of the trials, we added a pulse of 0.1 s duration and amplitude 10 cm/s on the velocity output of the model, either in the $x$ or $y$ direction. This pulse occurred randomly between 0.2 and 1.9 s after trial onset, to mimic perturbations at various movement periods.

After training, we tested the model on a centre-out-reaching task with eight targets equally distributed on a circle of 5-cm radius. Networks were subject to a variable delay period of between 1.2 and 1.7 s.

To probe online feedback correction and motor adaptation, we introduced a visuomotor rotation (VR) perturbation that rotated the output of the model by 30°, similar to previous visuomotor rotation experiments in humans[2] and monkeys[51]. For the simulations in Fig. 4G, this angle was increased to 45°.

## Neural recordings from behaving monkeys

We reanalysed previously published data from two macaque monkeys performing a VR adaptation task with a cursor controlled by movements of a manipulandum[51]. All surgical and behavioural procedures were approved by the Institutional Animal Care and Use Committee at Northwestern University (Chicago, USA).

In each session, the monkeys performed 154–217 successful trials to eight equally spaced targets. After this baseline period, a 30° rotation (clockwise or counterclockwise, depending on the session) of the cursor position feedback presented on a screen was introduced. Finally, after 219–316 successful adaptation trials, the perturbation was removed in order to study de-adaptation during this "washout" period. We quantified trial-by-trial learning by examining the monkey's hand trajectories, which were tracked by recording the position of the handle of the manipulandum.

We analysed the activity of populations of putative single neurons that were identified using standard sorting techniques and subsequent manual curation; data were recorded using two 96-channel micro-electrode arrays chronically implanted in the arm area of the primary motor cortex (M1) or dorsal premotor cortex (PMd; details in ref. 51).

## Data analysis

**Movement error metrics to quantify learning.** The take-off angle was defined as the initial reach direction, calculated between the go cue and peak velocity. When pooling the angular error across monkeys in Fig. 2, we smoothed the mean across all sessions from both animals using a Gaussian filter with s.d. of ten trials. When studying how error magnitude influences learning in the next trial (Fig. 5A), we computed the Pearson's correlation (*pearsonr* from `scipy.stats` package) between the absolute value of the angular error and the difference in absolute angular error between the current trial and the next trial. To assess whether these correlations were significant, we compared them to a null distribution under the assumption of joint normality.

**Analysis of temporally dissociable adaptation-related activity changes.** We sought to identify a "neural signature" of adaptation-related activity changes in the network that could be observed in neural recordings from monkeys performing the same VR adaptation task that we simulated. To this end, we probed three different behavioural epochs as follows (Fig. 3A). For the model data, we simulated 200 baseline trials (epoch A in Fig. 3A), 200 trials beginning immediately after perturbation onset (prior to any learning; epoch B), and 200 trials beginning 300 trials after the onset of learning (epoch C). For the monkey data, we considered the following epochs: 100 baseline trials (epoch A), the first 100 trials after perturbation onset, during which monkeys were beginning to adapt (epoch B), and the last 100

perturbation trials, when monkeys had learnt to counteract the perturbation (epoch C). Note that for the monkey data, the feedback epoch B was not as clearly defined as for the simulation data, since the monkeys had already started learning within epoch B—in fact, humans start learning after the first error[4]—and we thus could not isolate purely feedback-related activity changes.

The activity changes in the model was calculated by measuring, for each unit, the activity difference between all pairs of behavioural epochs (A, B, C in Fig. 3A). For this, we simulated the same trials (using the same random seed) without perturbations (A), with perturbations (B), and with perturbations after the network had adapted (C). To identify the time point within a trial at which the largest activity change happened, we computed the absolute value of activity change, and averaged the respective differences across neurons and trials. This resulted in the time courses shown in Fig. 3C. For the monkey data (Fig. 3D), since we could not have the exact trials in different epochs, we calculated the difference between trials in different epochs matched by the target in an all-to-all fashion, then averaged over those trial pairs.

After an initial analysis of the average activity change across all ten RNN models, we could define a "feedforward" time point (0.5 s after the go cue), in which the largest activity change between late adaptation (epoch C) and early adaptation (epoch B) happened, and a "feedback" time point (0.8 s after the go cue), in which the largest activity change between early adaptation (epoch B) and baseline (epoch A) happened. These values were very similar to those identified in the analysis of neural recordings from monkey M1, for which the feedforward time point happened 0.4 s after the go cue, and the feedback time point, 0.8 s after the go cue. For the pooled analysis presented in Fig. 3E, F and Supplementary Fig. 8E, F, we took the values of the activity change traces at those time points and calculated the ratio between the value at the feedforward time point and the value at the feedback time point.

**Assessing changes in the relationship between network dynamics and motor output during learning.** We trained a linear regression decoder to predict velocity from ongoing network activity (Fig. 4E, F). To mimic experimental studies in which recordings are only available from a subset of neurons, we only used 100 units for the analysis. For Fig. 4E, we predicted the time course of the cursor velocity subject to different time lags. We first did this during the baseline using non-overlapping sets of 100 training and testing trials, to establish the validity of our approach. Then, to assess the changes in the relationship between network dynamics and motor output following learning (Fig. 4F), we trained decoders on the last 100 trials of the adaptation phase, and tested them on each previous adaptation trial separately (similar to ref. 51).

**Analysis of learning timescales.** We investigated whether our model's learning time course is composed of two processes with different timescales by implementing the analysis used in earlier studies[71,72]. We used the following dual-rate state-space model, which we fitted to the angular error data:

$$x_f(n+1) = A_f x_f(n) + B_f e(n) \qquad (12)$$

$$x_s(n+1) = A_s x_s(n) + B_s e(n) \qquad (13)$$

$$x(n) = x_f(n) + x_s(n) \qquad (14)$$

The model was subject to the constraints $A_f < A_s$, $B_f > B_s$. $A_f$ and $B_f$ are the parameters describing the fast adaptation process, whereas $A_s$ and $B_s$ are the parameters describing the slow adaptation process. The *Adaptation* variable, $x$ (cf. Fig. 5B), was defined as the amount of

change in the take-off reaching direction, and the error, $e$, was defined as the take-off angular error scaled to [-1,1]. The four parameters $A_f$, $A_s$, $B_f$, $B_s$ were obtained by fitting the model to the adaptation time course observed in the simulated data using the *Sequential Least Squares Programming* method from the `scipy` optimisation package. We performed an *F-test* to test whether the dual-rate state-space model fits the adaptation data of our networks significantly better compared to a single-rate state-space model defined below:

$$x(n+1) = Ax(n) + Be(n) \qquad (15)$$

In Fig. 5C and Supplementary Fig. 9B, C, we illustrate the fit parameters of the dual-rate state-space model, using two different colours to indicate whether the dual-rate model fitted the simulated adaptation data significantly better than the single-rate model or not.

**Evaluating the impact of disrupted network dynamics on feedback-based adaptation.** In Fig. 5G–I and Supplementary Fig. 9D–G, we replicated and extended the experiments in ref. 44, in which the authors used subthreshold intracortical microstimulation during the instructed delay period to impair learning. In 50% of the trials between adaptation trial 10 and 50, we simulated a perturbation of the network activity by adding a small extra input to all the units in the network. The amplitude of this input was 0.1, and had a duration of 200 ms. In separate sets of simulations, we repeatedly delivered this perturbation at one of four different time windows during a trial (*Target*: 0–200 ms after trial start; *Go cue*: −200–0 ms before go signal; *Movement onset*: 400–600 ms after go signal; *Feedback*: 650–850 ms after go signal).

**Reporting summary**
Further information on research design is available in the Nature Portfolio Reporting Summary linked to this article.

## Data availability
The data that support the findings in this study are available from the corresponding authors upon request. Source data are provided with this paper.

## Code availability
All code to reproduce the main simulation results will be made freely available upon publication on GitHub (https://github.com/babaf/feedback-driven-plasticity).

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

## Acknowledgements

We thank Samuel D. McDougle and Adrian Haith for discussions about this research. M.G.P. acknowledges the Fonds de recherche du Quebec Santé (grant J1 chercheurs-boursiers en intelligence artificielle). L.E.M. received funding from the NIH National Institute of Neurological Disorders and Stroke (NS053603 and NS074044). C.C. received funding from the BBSRC (BB/N013956/1 and BB/N019008/1), the EPSRC (EP/R035806/1), the Wellcome Trust (200790/Z/16/Z) and Simons Foundation (564408). J.A.G. received funding from the EPSRC (EP/T020970/1) and the European Research Council (ERC-2020-StG-949660). The funders had no role in study design, data collection and analysis, decision to publish or preparation of the manuscript.

## Author contributions

B.F., C.C. and J.A.G. devised the project, interpreted the data and wrote the manuscript. M.G.P. and L.E.M. provided the monkey datasets. B.F. ran simulations, analysed data and generated figures. All authors discussed and edited the manuscript. C.C. and J.A.G. jointly supervised the work.

## Competing interests

J.A.G. receives funding from Meta Platform Technologies, LLC and Inbrain Neuroelectronics. The remaining authors declare no competing interests.
