## [Transparent Peer Review file · Nature Communications]

A neural implementation model of feedback-based motor learning

Corresponding Author: Dr Juan Alvaro Gallego

Version 0:

Reviewer comments:

Reviewer #1

(Remarks to the Author)

The authors present a model based on artificial neural networks which receives feedback about movement errors to update the synaptic weights as a model of M1 plasticity in sensorimotor adaptation tasks. The approach is interesting but the paper in its present form did not convince me for the following reasons, which are developed further below: (i) it is not clear what was the novelty of the study with respect to the training algorithm insofar as feedback can simply be considered as another input to a standard RNN model; (ii) the system model and control is very far from current models of human sensorimotor control, clearly there is a mismatch between the model and what is known about movement control. That the present simulations generalise to sensorimotor control is highly speculative, so is the link between the training algorithm and M1 plasticity. Finally (iii) the paper shows that some features emerge from the model such as fast and slow processes, without providing explanations, so it remains a black-box model which does not offer insight into the nature of the underlying phenomenon.

The biggest problem in my view is that the claims of novelty are based on a feedback-driven plasticity rule. From the perspective of sensorimotor adaptation, the fact that feedback drives adaptation is obvious. From the perspective of RNN, the equation did not clarify this point: how does that differ from a network that has additional input being the real-time trajectory? I believe that the novelty should be better stressed: is the update rule (page 12) a variant of back propagation or is similar? How is it derived and how does it contrast with previous models? There is a number of inaccuracies that made the reading of the mathematical contents difficult: in the first system of equations the error is indexed by the trial number (k), whereas in the activities dynamics it is summed over the neurons (index i). If the same error is broadcasted to all units, then the update rule seems like a straightforward backpropagation rule. If it is not the case it should be derived in detail. The update rule itself was unclear: now the error is summed over x and y which I assumed corresponded to neurons and trials, meaning that updating the weight (i,j) at time step t was based on error at that time from all trials. This is certainly unrealistic.

The packaging suggests that the proposed model is relevant to understand sensorimotor adaptation tasks in general but the system and adaptation rule was overly simplistic. The input/output corresponded to position/velocity is a fair simplifying approximation, but then it requires imagination to conclude that the model is suitable to capture higher dimensional state-space models (position, velocity, force, higher order derivatives) and associated learning rules that have yet to be characterized. The generalisability of the model and a more thorough discussion of how it could be applied to adaptation in more biologically plausible state-space control models was lacking (e.g. Izawa et al., 2008, *JNeurosci*, 28 (11), 2883-2891). Notably it is not clear that the distance to a straight path (equivalent to trajectory control) is the error signal that drives adaptation, thus there is a risk that reproducing features of neural activity with an implausible biological control model simply results from a spurious correspondence between artificial and biological systems.

The results presented in Fig. 4 are not very convincing: since it is an artificial system, we should be able to know exactly where the two timescales come from if we want to avoid accepting these results as a black-box, which offers no deeper understanding. Notably the black points in panel D for Af are very close to 1 which is higher than in data and seems to contradict the fit in panel C. There are known sensitivities of the dual-rate fitting procedure such that several sets of values can explain data equally well and I would recommend that the authors make more efforts to demonstrate that the process really contains two timescales with model selection techniques. Then by performing computer experiments it should be

possible to determine where the dual rate behaviour, if confirmed, comes from.

Reviewer #2

(Remarks to the Author)

Feulner et al. review

I want to start by saying that this is a well-written paper on a topic that is both near and dear to me and quite important. The neural basis of adaptation has been studied by many people for a very long time, but with few exceptions (some especially beautiful work on learning by a subset of the authors, some related NHP work by the Shenoy group, and some very nice studies in humans and animals by Shadmehr, Krakauer, and colleagues) most of the work has not resulted in a concrete theory about how an actual network of biological neurons might implement and facilitate adaptation. To that end, the authors here are doing exactly the right and important thing by going after this question using artificial networks where they can embody and test a particular hypothesis and mechanism. As enthusiastic as I am about this approach, I'm not sure if the authors have really paid off their hard work. Perhaps I'm missing something here, and if so, I sincerely apologize. I hope my questions and comments below will help the authors refocus and clarify their key contributions.

What is a plausible alternative hypothesis that the authors have in mind? I agree completely with the authors that the neural basis of adaptation is not yet fully known. However, most prior research is in agreement that motor adaptation involves an internal model with inputs corresponding to – at minimum – the feedback error signal during movement. For example, Lines 184-190 is exactly the current thinking in the field; are the authors suggesting that these two processes are new findings by them? VR adaptation is also often referred to as “error driven motor adaptation” for the precise reason that feedback error-signals are thought to be the primary drivers of learning.

What would be novel and quite surprising is if the authors proposed that the error signal was specifically computed by M1, and/or PMd & M1 were the key nodes for the internal model. But the authors in lines 315-318 state that the Cerebellum could easily be involved and since their network model is a single compartment, it is not currently possible to posit theories about what part of the learning computation M1 participates in. Finally, Line 79 at the end of the introduction, for example, is certainly the null hypothesis in the field or error-driven motor learning, where the precise nature of feedback that matters for the VR task is the positional error signal generated during movement.

On a more positive note, I'm generally quite happy with the modeling work that the authors have done. Certainly, I agree with their assertion in Lines 298-300 that such a model could be useful for testing very specific motifs embodied within OFC onto populations of neurons. At the same time, I found myself wanting the authors to do precisely that. In some ways, I think of the feedback error signal driving adaptation as a “sanity check” in their model and not a new finding. In Figure S4, for example, the authors train a network with SGD without overt feedback and find that a second peak in the activity change is absent. This is quite satisfying to see but, in my opinion, the expectation.

Perhaps the primary contribution lies in the model and how it may serve as a tool for performing future neuroscience? I seriously believe the modeling exercise is very well-done and important. For example, I can see high value in using the authors' trained model to test and generate hypotheses before designing new experiments in animals or humans. This accords with the huge efforts the authors have put into making sure the model behaves as expected and matches human and non-human primate measurements. If the authors wish to go this route, they can make even more direct comparisons between the monkey M1 data and their model, including showing that the dominant neural dimensions as well as well the characteristic motifs (e.g., the Churchland & Shenoy “prepare trigger execute” motif) during reaching are largely conserved across monkey and model.

Minor grammar issues:

Line 111: “network’s ability.... <missing word TO> ... flexibly...”

Line 117: “end of the line there are two “had”

Line 124: “learned” not “learnt”

Reviewer #3

(Remarks to the Author)

A widely accepted assumption of motor adaptation involves a combination of the “forward internal model,” which predicts the sensory consequences of ongoing actions, and the “feedback-driven process” of error minimization. However, the authors of this study argue against the necessity of forward internal models, claiming that the error-minimization computation alone can accomplish motor adaptation in visuomotor rotation tasks. This indicates that predicting sensory consequences is unnecessary, and error minimization is a sufficient computation.

To support their claim, the authors used a recurrent neural network (RNN) to generate the desired movement in a center-out-reaching task and adapt like its biological counterpart when the visuomotor rotation occurs. They compared the feedforward and feedback-related internal dynamics of RNN units and biological circuits to prove that error minimization is sufficient. Their RNN suggested two critical components in motor adaptation: 1) the error between the target and actual positions as an objective function and 2) a biologically plausible learning rule. The authors then demonstrated that their RNN could explain multiple motor adaptation phenomena reported in previous literature.

While this manuscript is very interesting and raises a fundamental question about the necessity of the forward internal model, it requires improvement in multiple aspects to claim sufficiency. The following concerns are divided into major and minor aspects, but they are not prioritized.

Major comments

1. The authors of this study delineated their logic by dichotomizing the process into error minimization versus prediction of sensory consequences and claiming the sufficiency of error minimization. To claim that error minimization is sufficient by developing a model, the authors' model may need to be as good as an alternative model that combines the prediction of sensory consequence with error minimization computation. If alternative models can explain human behavior significantly better, both qualitatively and quantitatively, the sufficiency claim may not stand.

However, the author's manuscript does not show the performance of an alternative model that includes both predictions of sensory consequence and error minimization computation. The authors only demonstrate the performance of a model where error minimization computation is removed. It is unclear whether readers should consider this error minimization removal model as a proxy for the model with "error minimization combined with the prediction of sensory consequence." If this was the authors' intention, it would be helpful to draw the linkage.

Furthermore, it is not clear where the model's prediction of sensory consequences occurs. Is it used as part of an objective function like error minimization (i.e., minimizing prediction error of sensory consequence or something similar)?

2. In Figure 1G, what is the objective function of the RNN model without error feedback? The method section provides an equation of the loss function for Figure 1F, but it is unclear what the objective function is for Figure 1G. Assuming the loss is the same equation except for the error signal (ϵ), the model's objective function for Figure 1G becomes "minimizing RNN unit activity, weights, and biases." This comparison seems to be between a supervised learning model (Figure 1F) and an unsupervised learning model (Figure 1G), and supervised learning with additional information would perform better than the other.

From here, a philosophical question and a practical question arise. The philosophical question is, "Does the machinery become a pure unsupervised learning machine without teacher signals once the machinery is not taking error signals into account?" The practical question is, "Can't the prediction of sensory consequence and/or any other information (e.g., a discrepancy with x, y-coordinate of the targets) be an objective function to generate training signal (besides error as teacher signal as the referred) instead of fully unsupervised?"

3. The authors argue that their RNN exhibits a dual-rate process similar to that of humans, but their claim is currently unconvincing for two reasons. Firstly, the parameters describing the dual-rate process seem incomparable to those from a human VR adaptation study (Figure 4D). Secondly, the authors never compared their RNN model with an alternative model (i.e., predicting sensory consequence + error minimization model).

Regarding the first point, the difference in retention factors between A_f and A_s appears negligible in their results, unlike McDougale et al. Moreover, B_s generally seem larger than human results, and B_f and B_s largely overlap around 0.1-0.2 (although a statistical test is needed to confirm this). This raises two questions: 1) Would a model with a single parameter B_{avg} (a non-multiple timescale model) perform significantly worse than the dual-rate model for the network-generated trajectory (estimated by AIC or any other model comparison value)? 2) What if the model with sensory consequence prediction combined with error minimization (not shown in the manuscript) is more compatible with the dual-rate process than the model with only error minimization?

Without addressing the first question, it would be difficult to claim that the "model exhibits dual-rate like a human" since A_f and A_s do not appear to differ, and B_{avg} could be as good as having two different B_s . If adding the feedforward internal model generates dual-rate behavior better than the error minimization model, then the authors' fundamental claim that error minimization is sufficient would not hold.

4. In a similar direction, the variance of panels in figures 2C and 2F does not seem compatible (in which the model's variance is visibly smaller). If the model with a feedforward internal model (i.e., prediction of sensory consequences + error minimization) can capture the variance pattern, would the author's sufficiency claim stand?

Minor comments

1. The authors clearly summarized the input and output in figure 1B. However, I'm curious about the input and output at each time step for the RNN. Based on my understanding, the input includes velocity and error at time t , while the output is the velocity at time $t+1$. Does the RNN also compute the prediction of sensory consequences, such as error calculation, and feed it back into the network?

2. I noticed a hold signal in figure 1B, but I don't know if it continues to influence the movements once the RNN begins generating them. How do the authors negate the hold signal after the initial moment of each trial?

3. What would be the role of random velocity bump perturbation? Why was it added to the RNN training procedure? What happens if this procedure is removed?

4. How is the eligibility trace $r(t)$ calculated in line 391? Is the $r(t)$ different between the equation in line 366 and the equation in line 391?

5. What is the equation of the gradient-based learning rule in contrast to the local plasticity rule? Is it identical to the models in Figures 1 and 2?

6. Like the claim above, would the performance stay the same if the weight update rule includes the $F_{\text{predict_sensory}}$ matrix additively (or multiplicatively)?
7. In figure 3E, the authors included a model with 20% of neurons receiving feedback to match the FB/FF ratio with macaque (as shown in figure S4B). However, the manuscript notes that 73% of biological neurons have feedback responses. If the model includes a prediction of sensory consequence component, would the discrepancy decrease when 80% of model neurons receive feedback?
8. Is the data in figure 4B human result or RNN result? It would be helpful to appreciate the result by putting the outcome of humans and RNN side-by-side. The same applies to figure 4F.
9. It would be helpful to provide the rationale for the learning sequence for the 1100 divided epochs.
10. Lastly, I attempted to access the authors' GitHub repository for more model details but was unsuccessful. Could you provide access to it?

Reviewer #4

(Remarks to the Author)

The authors conducted an experiment to test their hypothesis that error signals for online feedback control and local plasticity rules are sufficient to induce trial-by-trial motor adaptation to the visuomotor rotation observed in both monkeys and humans. To this end, they developed an RNN model that employed error-based learning and online feedback control using identical error feedback signals. The simulation results showed that the RNNs achieved feedback motor control and motor learning at the same time by utilizing the same error-feedback signals. Moreover, the activity changes during learning demonstrated distinct feedback and feedforward components, consistent with the observations of monkeys' M1 neuronal activity. Finally, the RNN model reproduced the characteristic behavioural features observed in monkeys and humans. Based on these findings, the authors concluded that trial-by-trial motor adaptation can be achieved by directly updating the feedback control policy with a local plasticity rule, without the need for a forward model.

This study provides valuable insights into two longstanding questions in motor control research: whether updating the forward model is essential for motor adaptation, and how motor control and motor learning can be achieved with the same circuit. However, I believe that additional discussions and analyses of the biological plausibility and similarity to experimental data would greatly increase the significance of their model as a biological model of animal motor learning.

Major comments:

1) The biological plausibility of the local plasticity rule

- a. The present model assumes that the error feedback signal is provided as one of the presynaptic inputs, rather than a separate teacher signal (e.g., climbing fiber input to Purkinje cells). However, it is not entirely clear how the postsynaptic neurons would be able to distinguish the error input among the numerous other inputs and use it effectively as a teacher signal. This is an important question, as it is not clear how such a distinction could be made in a real neural network.
- b. In addition, the model assumes that only the internal connections (W) are updated, but it is unclear how these connections can be distinguished from other (W_{in} or F) inputs projected to the same postsynaptic neuron.
- c. Fig. 3 suggests that error signals can be distinguished from other signals by their timing within a trial: error FB input comes later (~700ms after go cue) in the trial than other FF input (~500ms). However, it raises questions about the biological plausibility of the LTP mechanism when the error input is delayed by approximately 200 ms. This is because LTP is typically thought to occur within a more precise time window, on the order of tens of milliseconds, in the case of NMDA receptor-induced spike-timing dependent plasticity [1]. The authors could provide further clarification on this point to better explain the biophysical plausibility of their proposed mechanism.
[1] Caporale, N. & Dan, Y. Spike Timing-Dependent Plasticity: A Hebbian Learning Rule. *Annu Rev Neurosci* 31, 25–46 (2008).

2) Neural basis for the feedforward motor control

The authors posit that learning induces circuits in M1 to produce feedforward motor commands, while the feedback error learning model proposed by Kawato et al. postulates that the inverse model is acquired in the cerebellum [2] (as cited). The cerebellum fulfills the prerequisites for error-based plasticity highlighted in my previous comment, including the presence of an independent error signal (climbing fiber input) and a relatively long LTD time window (~200 ms). As such, Kawato et al. suggest that the cerebellum may develop an inverse model for generating feedforward motor commands. The differences between the feedback error learning model of Kawato et al. and the present study's approach, as well as the potential for the cerebellum's inverse model to produce feedforward motor commands, warrant further discussion.

[2] Kawato, M. & Gomi, H. A computational model of four regions of the cerebellum based on feedback-error learning. *Biological cybernetics* 68, 95–103 (1992).

3) Relationship with behavioural data

- a. While it is impressive that the model can replicate numerous behavioural features from previous studies, further elaboration on what are the specific features of the model critical to reproducing each behavioural feature would be beneficial.

b. The authors suggest that the 2-state model can account for the current results; however, it is not entirely clear whether the analysis truly supports the existence of two distinct states (fast and slow states). An essential finding in the previous studies on the 2-state model is the occurrence of spontaneous recovery after de-adaptation (Smith et al. 2015). To lend support to the 2-state model, the authors should conduct an additional simulation to determine whether spontaneous recovery is replicated with their model.

c. In the discussion, the authors associate each learning process (fast/slow) to two neural activity components (FB/FF). However, the association between the learning processes and neural activity components is not straightforward for me. The adaptation curve in Fig. 4C measures the take-off error, which, in principle, does not reflect the feedback response. Thus, it is unclear how the feedback neural response could influence the fast learning component. The authors should provide more detailed elaboration on the association between learning processes and neural activity components to clarify this issue.

4) The relationship between the neural activity components of FB and FF and learning

a. If I understand correctly, it would be beneficial if the authors could provide an explanation for why the monkey M1 neural activity data were not analyzed in a similar manner to the model to elucidate the FF and FB components (i.e. B-A and C-A). As these components are expected to be present in the monkey data, their analysis would provide further insight into the neural mechanisms underlying motor learning.

b. The proposed idea is that the FB component in one trial influences the synaptic weights and leads to the appearance of the FF component in the next trial. If this is indeed the case, there should be a correlation between the FB component and the FF component of the next trial, on a trial-by-trial basis. Although the authors have demonstrated trial-by-trial learning in terms of the behavioural correlation between error size and learning size (Fig. 4A), it would be highly supportive if they could demonstrate a trial-by-trial shift from FB to FF at the level of neural activity. If demonstrating the correlation between the FF and FB components is challenging due to inherent noise in the neural signal, illustrating the adaptation curve for these components can still provide more insights."

Minor comments:

Line 51: The authors cited Kawato's studies on feedback error learning in the context of the forward model. However, these studies mainly focus on the training of the inverse model in the cerebellum. To better support the forward model-based learning, it may be beneficial to cite references such as Shadmehr et al. (2010) and Flanagan et al. (2003), which emphasize the role of prediction in motor learning.

Shadmehr R, Smith MA, Krakauer JW (2010) Error correction, sensory prediction, and adaptation in motor control. *Annual review of neuroscience* 33:89–108.

Flanagan JR, Vetter P, Johansson RS, Wolpert DM (2003) Prediction Precedes Control in Motor Learning. *Curr Biol* 13:146–150

Line 199-200 and the legend of Fig. 3C: it would be helpful to indicate that the activity change was measured as an absolute value.

Line 249: Albert et al demonstrated the temporal correlation with a time shift between the feedback response and the feedforward activity in the next trial. Did the authors observe a similar temporal correlation in the RNN and monkey M1 activity?

Fig. 4B, it would be helpful to provide an explanation as to why they used Movement error instead of the takeoff error, which was used in other sections of the study.

Line 392: the authors should add an explanation of why they summed every fifth time step.

Line 401: "10/s". Does this mean "10 cm/s"?

Line 440: for the analysis of Fig. 4A, the authors correlated "the absolute value of the angular error" and "the (signed) difference in the angular error," but it is not clear why this always resulted in a positive correlation. Did the authors use the absolute value for both parameters?

Version 1:

Reviewer comments:

Reviewer #1

(Remarks to the Author)

The authors responded to most of my comments but I remain doubtful about the originality of the proposed learning rule and the added section, I admit, made it even more confusing. Backpropagation is gradient descent from a loss function with a step size. Here, the learning rule differs from backpropagation because the gradient of the loss function is not used to update the weights. But in this case it corresponds to a different loss, and I was missing a theoretical argument on why it works, or how to justify it from an optimization theory perspective. Does it minimize the loss function defined on page 14? If so, does that mean it's approximately aligned with the gradient? If not, what loss function does it minimize? Or is the update rule

proposed by the authors only heuristic and we should just cross our fingers and hope it works? I recommend publication provided that the authors can clarify and rework the mathematical basis of the proposed rule.

(Remarks on code availability)

Reviewer #2

(Remarks to the Author)

I want to thank the authors for writing this very interesting manuscript. I feel that the authors have done a very commendable job in responding to all of my previous comments. There are a number of new analyses to further explore the mechanism of their model. They have also done a lot of work on re-wording and/or clarifying various aspects of this manuscript. I feel that my most pressing concerns have now been addressed, and any further back-and-forth (between me and the authors) would only function to quench my personal remaining curiosity rather than improve the communication of the science in a meaningful way. I want to once again thank the authors for this nice work.

(Remarks on code availability)

Reviewer #3

(Remarks to the Author)

The reviewer is deeply grateful for the authors' diligent efforts in making the manuscript clearer and more comprehensible. The reviewer also highly values the invaluable guidance provided by the code they submitted, which significantly enhanced the understanding of the manuscript.

The reviewer thought they did not implement the feedforward component; however, the authors expected it to emerge, which I missed. Overall, the current version clearly demonstrates that the authors are not arguing against the necessity of a feedforward controller but are emphasizing that the recurrent connection could implicitly implement one.

The reviewer has minor comments.

1. I would like to believe the result in Figure 4 indicates intermixed neurons for feedback and recurrent activity, which corresponds to Figures 4A-C. However, I could see separate clouds in the lower left corner of Figure 4D (and some in Figures S2B and C). Do the authors mind if they show some numerical results for clustering analysis in the supplementary result? Otherwise, can the readers consider it trivial noise?

2. Among many new analyses and figures, the new supplementary figure 1 is extremely helpful in understanding the paper's core message and linkage to recent literature. Would the authors consider incorporating that somewhere in the main figure?

3. Although the reviewer could follow the logic of the model by skimming through the well-written code, the model was not trained when I ran the attached Zipcode. The README was great, and the codebase was clear, but the VR perturbation code and automatized experiment code resulted in errors.

Besides these minor comments, the reviewer thinks clarification of key assumptions and new results support their main claim.

(Remarks on code availability)

The README file attached to the codebase helped me run the well-written code. According to the instructions, I ran the code and tested whether it generated results similar to the manuscript. Although setup-related codebases (setup_parameters.py, setup_OFC_network.py) worked fine, the VR rotation code (adaptation_learning.py) and automatized experiment code (paper.py) did not work and kept providing me NaN as the outcome.

Reviewer #4

(Remarks to the Author)

First of all, I would like to respect the author's efforts, such as adding the experiment. However, I believe that some improvements need to be considered to demonstrate the significance of the study more clearly.

Major comments

Comment #1 Introduction is still confusing

The introduction contains multiple topics of research mixed together, making it difficult for the reader to understand what the main issue is. I find that at least two major topics are described: the first is whether the internal forward model is necessary for motor adaptation, and the second is whether the feedforward and feedback controls are implemented independently. I recommend that the authors explicitly separate these two topics and clarify the relationship between the two topics and their relevance to the current study. Particularly confusing is that there is no "forward model" in Fig. S1, the conceptual diagram of their hypothesis. Therefore, it is still unclear how these two topics (forward model and feedforward controller (expressed as an "internal model" in Fig. S1)) are connected to the current study.

Comment #2 Evidence for independence of neural mechanisms of feedback-based correction and adaptation is not clear

(2-1) It is stated that RNN units involved in feedback-based correction and adaptation are not separated because the distribution of the recurrent and feedback weights (W and F) showed only one cluster. However, Fig. 4D shows that there is another cluster near the zero of the horizontal axis (= weight F). It is necessary to have objective reasoning to claim that this distribution is only a "single large cluster".

(2-2) I assume that the analysis of the weights in Fig. 4D is done with a model trained only on the recurrent connection (W) (this is also not clearly written, though). However, if I understand correctly, the weight F is arbitrarily initialized in the model. Since the authors must also have models with trained input and feedback weights (W_{in} and F), I suggest examining the independence of the modules in these networks.

(2-3) Basically, the recurrent weights (W) and feedback weights (F) represent synaptic connections and do not indicate how the units are actually activated during the task. In order to directly examine the separation of functional modules, they should examine whether units with feedback (FB) and feedforward (FF) activity components, as examined in Fig. 3C, are represented in independent clusters of neurons or not.

(2-4) Finally, it was not shown whether neurons involved in feedback-based correction and adaptation also overlap in the monkey motor cortex. Although it is difficult to show synaptic weights (Fig. 4D) in animals, it is possible to show whether neurons with significant FB and FF activity changes (Fig. 3C) overlap or not.

Comment #3 The significance of the new decoding analysis is not clear.

It is not clear why the ability to decode can be evidence for an update of the control policy, rather than other possibilities (e.g., an update of the forward model). I guess that even if neural activity in the motor cortex is involved in feedback or feedforward control, it is possible that neural activity in the motor cortex correlates with current or future hand velocity, depending on the presence of noise or external perturbation.

Comment #4 Results do not seem to support the dual-state model

(4-1) First, I could not see why the fast and slow parameters (A_{fast} and A_{slow} , B_{fast} and B_{slow}) existed even when the best fit was achieved with the one-state model in Fig. 5C.

(4-2) Generally speaking, using more model parameters should increase fitting accuracy, so it is not surprising that a dual-state model has better fitting accuracy.

(4-3) More critically, if spontaneous recovery is not seen, it is reasonable to assume that there are no dual-state models. I am not pointing out this result negatively, but the limitations of this present model should also be fairly described and discussed in the manuscript (not only for responses for reviewers) because such discrepancies between the experimental data and the model provide important insights for future studies.

Minor comments

Line 216: Is it a "unit" instead of a "neuron"?

(Remarks on code availability)

Response to reviewers' comments Feulner *et al.* "Feedback-based motor control can guide plasticity and drive rapid learning"

Reviewer 1

The authors present a model based on artificial neural networks which receives feedback about movement errors to update the synaptic weights as a model of M1 plasticity in sensorimotor adaptation tasks. The approach is interesting but the paper in its present form did not convince me for the following reasons, which are developed further below: (i) it is not clear what was the novelty of the study with respect to the training algorithm insofar as feedback can simply be considered as another input to a standard RNN model; (ii) the system model and control is very far from current models of human sensorimotor control, clearly there is a mismatch between the model and what is known about movement control. That the present simulations generalise to sensorimotor control is highly speculative, so is the link between the training algorithm and M1 plasticity. Finally (iii) the paper shows that some features emerge from the model such as fast and slow processes, without providing explanations, so it remains a black-box model which does not offer insight into the nature of the underlying phenomenon.

We are glad the reviewer finds our paper interesting, and appreciate their comments, which we have carefully addressed point by point below. In particular, we have added as many as 6 analyses and/or simulations and numerous clarifications that we expect to address all of the reviewer's concerns. Regarding their three main points:

(i) Our learning rule is both conceptually and practically very different from adding another input to a standard RNN model. First, in contrast to classic RNN models of the motor system, the network does not simply produce an output pattern based on an input cue, it *controls it in real-time* based on the (error) feedback it receives. Second, we show that feedback is not only sufficient for control but also for learning. Third, it is true that there are powerful optimisers such as backpropagation through time that make learning in recurrent networks straightforward. However, these optimisation methods are not biologically plausible because they use non-local information (both in time, and in space: synapses would need to "know" what other synapses, which are not locally connected, did long into the past). In contrast to these powerful optimisers, the few currently available biologically plausible learning rules for recurrent networks typically do not work well for relatively complicated tasks such as the paradigm we studied here. However, we show that by simply combining online feedback with a biologically plausible learning rule, learning can be achieved in a local, simple fashion.

(ii) We agree that both the model—a single recurrent circuit—and the plant—a "moving dot" that represents the end point position of the hand—are much simpler than their biological counterparts. However, our goal was to build the simplest possible model to address our hypothesis that the same recurrent circuit that performs (feedback-based) motor control can achieve trial-by-trial learning using a biologically plausible rule. Based on the reviewer's suggestions, we have added extra analyses and simulations to show that our results hold across a broad range of assumptions. We have also expanded on our discussion of future work to extend this study, which we regard as proof of concept.

(iii) We agree that the dual-process model was largely underexplored in the original manuscript. Following the reviewer's suggestions, we have added a new series of analyses and simulations that have allowed us to gain insight into the key underlying factor driving this observation.

1. The biggest problem in my view is that the claims of novelty are based on a feedback-driven plasticity rule. From the perspective of sensorimotor adaptation, the fact that feedback drives adaptation is obvious. From the perspective of RNN, the equation did not clarify this point: how does that differ from a network that has additional input being the real-time trajectory? I believe that the novelty should be better stressed: is the update rule (page 12) a variant of back propagation or is similar? How is it derived and how does it contrast with previous models? There is a number of inaccuracies that made the reading of the mathematical contents difficult: in the first system of

equations the error is indexed by the trial number (k), whereas in the activities dynamics it is summed over the neurons (index i). If the same error is broadcasted to all units, then the update rule seems like a straightforward backpropagation rule. If it is not the case it should be derived in detail. The update rule itself was unclear: now the error is summed over x and y which I assumed corresponded to neurons and trials, meaning that updating the weight (i,j) at time step t was based on error at that time from all trials. This is certainly unrealistic.

As outlined above, our learning rule is not related to backpropagation through time, it is a simpler, biologically plausible learning rule. Backpropagation through time is a very powerful method but it is not local (because it requires information about different timesteps and of other weights), therefore it is regarded as biologically implausible (Lillicrap *et al.*, 2020; Murray, 2019). Our learning rule is different from backpropagation in two fundamental ways: (1) errors are not backpropagated through time, and (2) feedback weights are not derived from the forward output weights, but instead learned during the initial training phase. Our learning rule is therefore local and biologically plausible.

To clarify these differences, we have added a new section in the Methods comparing our learning rule to standard gradient descent, and expanded the comparison on biologically plausible learning rules and their potential implementation in biological brains in the Discussion. We also clarified the mathematical content: the index (k) in the first system of equations represented the output dimensions x and y , not the trial number. It was unfortunate that we used (k) again for the update rule where it represented indeed the trial. We changed that now and added a sentence to prevent confusion in the future.

2. The packaging suggests that the proposed model is relevant to understand sensorimotor adaptation tasks in general but the system and adaptation rule was overly simplistic. The input/output corresponded to position/velocity is a fair simplifying approximation, but then it requires imagination to conclude that the model is suitable to capture higher dimensional state-space models (position, velocity, force, higher order derivatives) and associated learning rules that have yet to be characterized. The generalisability of the model and a more thorough discussion of how it could be applied to adaptation in more biologically plausible state-space control models was lacking (e.g. Izawa *et al.*, 2008, *JNeurosci*, 28 (11), 2883-2891). Notably it is not clear that the distance to a straight path (equivalent to trajectory control) is the error signal that drives adaptation, thus there is a risk that reproducing features of neural activity with an implausible biological control model simply results from a spurious correspondence between artificial and biological systems.

We are glad that the reviewer agrees that assuming position/velocity-based control and feedback is a fair approximation to study sensorimotor adaptation, and appreciate their suggestion to explore whether our findings would generalise to feedback signals other than distance to a straight path. We have now run simulations with both higher-dimensional feedback, and endpoint instead of trajectory control, to show that our findings extend beyond the assumptions made in the original manuscript.

In the first simulation, we extended our original model to have a higher dimensional state-space including both position and velocity variables and found that the model can still learn, and it does so with an “exponential” time course that resembles experimental observations (Fig. R1A-C). In the second simulation, we used endpoint location instead of trajectory control (Cluff & Scott, 2015) and showed that the model again learned in a similar way (Fig. R1D-F). Finally, we also tested what would happen in the presence of degraded sensory feedback, and found that noisier “sensory feedback” of different types impairs but does not preclude adaptation (Fig. R2), in an effect reminiscent of behavioural observations in cerebellar patients (Rabe *et al.*, 2009).

We have now added these five new sets of simulations to the paper as new Supplementary Figures 6 and 7 and discussed them where appropriate.

Figure R1. Online motor control and feedback-based learning can be achieved for both higher-dimensional joint position and velocity feedback (A-C), and for endpoint instead of trajectory control (D-F).

Figure R2. Control and adaptation can be achieved with noisy signals and less localised plasticity rule. A-C. Learning in the presence of additive input noise. D-F. Learning in the presence of velocity-dependent noise applied to the output. G-I. Learning in the presence of additive feedback noise.

3. The results presented in Fig. 4 are not very convincing: since it is an artificial system, we should be able to know exactly where the two timescales come from if we want to avoid accepting these results as a black-box, which offers no deeper understanding. Notably the black points in panel D for Af are very close to 1 which is higher than in data and seems to contradict the fit in panel C. There are known sensitivities of the dual-rate fitting procedure such that several sets of values can explain data equally well and I would recommend that the authors make more efforts to demonstrate that the process really contains two timescales with model selection techniques. Then by performing computer experiments it should be possible to determine where the dual rate behaviour, if confirmed, comes from.

We agree with the reviewer that the two timescales result was underexplored in the original paper. Inspired by this comment, we have now performed additional simulations to further characterise and understand this observation.

First, we tested whether a two-timescale model explained the learning process significantly better than a one-timescale model (Figure R3). We found that a two-timescale model fitted the data better only in a subset of networks (Figure R3B). We were intrigued by this discrepancy across models that only varied in parameter initialisation, and posited that it would be related to the nonlinear effects of recurrent connectivity on network activity and behaviour. To test this intuition, we performed a second set of simulations in which we limited the recurrent connectivity across units (from 100 % connection probability to 50 %). These less recurrently connected networks were more likely to be better explained by a single timescale model than fully recurrently connected networks (compare Figure R3B and R3C), indicating that a single timescale learning process was more likely to occur in networks with less recurrent connectivity. Finally, in addition to these new analyses and simulations, in the revised manuscript we also present a related analysis unveiling the interaction between feedback-related and feedforward-related components of learning in the model (new main Figure 4, new Supplementary Figure 3G-J), which are described under the new section “Recurrent neural networks learn through functionally distinct processes that are intermixed in their implementation.”

We have added the new comparison of one- and two-timescale learning process fits to our various models as part of revised Figure 5 and new Supplementary Figure 9.

Figure R3. Comparison of two timescale and one timescale model fits. **A.** Example model fits for one recurrently fully connected network. **B.** Parameter fits for ten networks differing in their initial seeds colour-coded based on the best fit (one process vs. two processes). Note the mixed results in terms of best model fit. **C.** Same as (B) but for networks with 50 % recurrent connectivity. Note how in this case almost all models are best described by an order one process.

Reviewer 2

I want to start by saying that this is a well-written paper on a topic that is both near and dear to me and quite important. The neural basis of adaptation has been studied by many people for a very long time, but with few exceptions (some especially beautiful work on learning by a subset of the authors, some related NHP work by the Shenoy group, and some very nice studies in humans and animals by Shadmehr, Krakauer, and colleagues) most of the work has not resulted in a concrete theory about how an actual network of biological neurons might implement and facilitate adaptation. To that end, the authors here are doing exactly the right and important thing by going after this question using artificial networks where they can embody and test a particular hypothesis and mechanism. As enthusiastic as I am about this approach, I'm not sure if the authors have really paid off their hard work. Perhaps I'm missing something here, and if so, I sincerely apologize. I hope my questions and comments below will help the authors refocus and clarify their key contributions.

We thank the reviewer for their enthusiasm about our work, and are glad that they think the paper is well written and addresses an important topic. Inspired by their comment and those by the other reviewers, we have now tried to articulate the key conceptual novelties of our paper better, which we expand on below as well as in the revised version of our manuscript. Briefly, these are:

- 1) We show that a recurrent neural network that can do feedback-based motor *control* can use the same feedback signal to *learn* to counteract a perturbation. This is both novel and significant because our networks learn in a biologically plausible way from a “mechanistic” (i.e., the learning rule is biologically plausible, the feedback too) point of view, show activity changes that are consistent to those observed in monkey M1, and readily reproduce up to six additional behavioural studies.
- 2) Our model provides computational support (and integrates) recent proposals based on behavioural studies that trial-by-trial error signals could be used as teaching signals for adaptation (Albert & Shadmehr, 2016), that feedback and feedforward control are intimately related (Maeda et al., 2020), and that motor adaptation can be achieved purely based on policy updates (Hadjiosif et al., 2021), thus contributing to an ongoing reformulation of long-held positions in the field.
- 3) Classic models of motor control and learning map the different “computational modules” underlying motor adaptation (control policy, forward model, state estimate, etc.) in an isomorphic manner to different brain regions (e.g., (Kawato et al., 1987; Shadmehr & Krakauer, 2008; Todorov & Jordan, 2002), although more recent evidence challenges this view, e.g., (Krakauer et al., 2019)). Here we show that all the necessary “computations” can emerge from the intrinsic properties of recurrent circuits that control a plant and are endowed with a local plasticity rule.

To clarify the key contributions we have thoroughly revised the text in the Abstract, Introduction as well as throughout the paper, and created a new conceptual schematic (Figure R4, added as new Supplementary Figure 1).

[figure redacted]

Figure R4. Comparison between a classic model of motor adaptation (A), and the model proposed in this manuscript (B).

1. What is a plausible alternative hypothesis that the authors have in mind? I agree completely with the authors that the neural basis of adaptation is not yet fully known. However, most prior research is in agreement that motor adaptation involves an internal model with inputs corresponding to – at minimum – the feedback error signal during movement. For example, Lines 184-190 is exactly the current thinking in the field; are the authors suggesting that these two processes are new findings by them? VR adaptation is also often referred to as “error driven motor adaptation” for the precise reason that feedback error-signals are thought to be the primary drivers of learning.

We agree with the reviewer that these ideas were not well articulated in the original manuscript, and hope that the key novelties —as outlined in the response immediately above— are clearer in the thoroughly revised version of the manuscript. For example, after reviewing key recent behavioural findings that inspired some of our assumptions, the Introduction now states:

Here, we hypothesised that a recurrent circuit that controls its output based on an incoming error signal can leverage this same feedback signal to learn to adapt to a persistent perturbation through targeted synaptic changes. Our hypothesis implies that all processes necessary for rapid learning can emerge from a single recurrent circuit performing error minimisation, rather than each being implemented by an independent module. It also assumes that updating an explicit forward internal model is not necessary for successful adaptation.

We believe that the fact that our model recapitulates as many as five (in the revised version) different behavioural studies and does so by generating activity patterns that match electrophysiological recordings from monkey primary motor cortex (as compared to monkey premotor cortex; see new Supplementary Figure 8D-F), supports our model’s proposal that the different computations necessary for sensorimotor adaptation can emerge from a recurrent circuit that controls its output and learns to update a control policy without updating an explicit feedforward internal model. This contrasts with previous accounts that mapped each computation into a different brain region.

Regarding Lines 184-190, we didn’t mean to suggest that this was a new finding, but rather that this was a way to validate that our model “behaves” according to what is expected to happen based on the existing literature. We have now rephrased the text and added the appropriate references.

We have modified the text throughout the manuscript to better articulate the findings that justify some of the assumptions in the model, the key novelties of this work, and how these relate to past work and can be extended in the future.

2. What would be novel and quite surprising is if the authors proposed that the error signal was specifically computed by M1, and/or PMd & M1 were the key nodes for the internal model. But the authors in lines 315-318 state that the Cerebellum could easily be involved and since their network model is a single compartment, it is not currently possible to posit theories about what part of the learning computation M1 participates in. Finally, Line 79 at the end of the introduction, for example, is certainly the null hypothesis in the field or error-driven motor learning, where the precise nature of feedback that matters for the VR task is the positional error signal generated during movement.

Thank you for raising these points. As mentioned above, one of the key findings in our work is that the different computations underlying motor adaptation need not each map one-to-one onto a specific region. As such, PMd and M1 can indeed be seen as nodes for a “distributed” internal model —or of the controller, etc. We have added several new analyses and simulations to more directly show this.

First, we have performed a new analysis that shows that, after initial training, networks developed a predictive (feedforward) model that predicted future movements from ongoing activity (Figure R5A,B). This model gets updated during adaptation as part of the learning process (Figure R5C), in a manner consistent with direct policy update (Hadjiosif *et al.*, 2021) based on a “teacher signal” that is derived from the ongoing error (Albert & Shadmehr, 2016). In parallel to this process, the network performs feedback-driven corrections, which dominate early during adaptation (Figure R5D,E). These two learning processes overlap in their implementation by single network units, as indicated by connection weight (Figure R5F) and single unit activity analyses (Figure R5G).

Figure R5. Characterisation of adaptation in the network model. **A.** Accuracy of decoder trained on predicting time-lagged velocity output. **B.** Same as A but for network without recurrent connectivity. **C.** Accuracy of decoder trained on the last 100 trials of adaptation degrades when tested in earlier trials indicating a learned policy shift. Single trial accuracy (grey) and smoothed with Gaussian filter of width 10 trials (black). **D.** Average activity change of all neurons in an example network. Activity change is measured as absolute difference. **E.** Time course of average activity change (D) for two fixed time windows within the trial at 500ms (green) and 800ms (blue) after go. **F.** Network connectivity before adaptation for example network. Single dots represent single units and their average feedback input (x-axis) and recurrent input (y-axis) weight. **G.** Activity change to respective baseline trials of example units during adaptation. Single units switch from showing feedback-related (activity change around 800ms after go) to showing learning related (activity change around 500ms after go) activity changes.

Second, we have performed another set of simulations that confirm that recurrent connectivity is necessary for feedback-based control with biologically plausible delays (Figure R6A-F), and showed that it does so by integrating incoming feedback (Figure R6G-J). Note that this is not to say that “everything is everywhere” in the brain. We expect to find differences across regions, as suggested by a new comparison between network activity changes and activity changes in monkey M1 and PMd (details in the response to Comment 4 below).

Figure R6. Recurrent connectivity enables stable control with delayed feedback. Three different model variants are compared in A-F. Left (A,D), a network without any recurrent connections and with no time delay in the feedback signal. Middle (B,E), the same network without recurrent connections but with a time delayed feedback signal. Right (C,F), the default model used throughout the paper with recurrent connections and a time delayed feedback signal. **A-C.** Hand trajectories produced by the RNN. **D-F.** Hand trajectories after introducing a 30° rotation of the RNN's output, to mimic a visuomotor rotation perturbation. **G.** Average recurrent (blue) and feedback input (red) as well as average feedback response (black). The average feedback response is computed by taking the difference in neural activity between perturbed and unperturbed trials. **H.** Feedback component against feedback input for single neurons 620ms after GO signal. **I.** Same as (J) but 800ms after GO signal. **J.** Same as (G) but for network without any recurrent connections.

We have added the new analyses and simulations characterising adaptation in the networks as a new main Figure 4 and a new section “Recurrent neural networks learn through functionally distinct processes that are intermixed in their implementation” in the paper. The simulations and analyses showcasing the necessity of processing incoming feedback through recurrent connectivity is shown in new Supplementary Figure 3. Besides, as mentioned above, we have modified the entire manuscript, with emphasis on the Introduction and Discussion, to elaborate on these ideas.

- On a more positive note, I'm generally quite happy with the modeling work that the authors have done. Certainly, I agree with their assertion in Lines 298-300 that such a model could be useful for testing very specific motifs embodied within OFC onto populations of neurons. At the same time, I found myself wanting the authors to do precisely that. In some ways, I think of the feedback error signal driving adaptation as a "sanity check" in their model and not a new finding. In Figure S4, for example, the authors train a network with SGD without overt feedback and find that a second peak in the activity change is absent. This is quite satisfying to see but, in my opinion, the expectation.

We are glad the reviewer found this analysis satisfying, and we have now added additional analyses and simulations to further understand learning in our model. We have complemented these analyses with additional simulations manipulating different aspects of the network to "causally" identify the functional contributions of different components.

Figure R7, partially reproduces Figure R2 above. Control and adaptation can be achieved with noisy signals and less localised plasticity rule. **A–C.** Learning in the presence of additive input noise. **D–F.** Learning in the presence of velocity-dependent noise

applied to the output. **G–I.** Learning in the presence of additive feedback noise. **G–I.** Learning in the presence of additive feedback noise. **J–L.** Learning with plasticity in all incoming weights.

The first set of new analyses and simulations was our detailed investigation of adaptation in the network, presented in Figure R5 and discussed above. In brief, this work confirmed that recurrent circuits that perform feedback-based control can use a biologically plausible plasticity rule to learn via direct policy update; all computations were intermingled in the implementation by network units since distinct modules did not emerge. Second, we also examined the effect that noise in different parts of the model feedback had on motor adaptation, as well as whether adaptation would be qualitatively different if feedback connections were also plastic (Figure R7). In all cases the network could still adapt to a visuomotor rotation perturbation.

We have added these additional analyses and simulations describing adaptation in our model and manipulating different components to report the functional effects on adaptation as new main Figure 4 and new Supplementary Figure 6.

4. Perhaps the primary contribution lies in the model and how it may serve as a tool for performing future neuroscience? I seriously believe the modeling exercise is very well-done and important. For example, I can see high value in using the authors' trained model to test and generate hypotheses before designing new experiments in animals or humans. This accords with the huge efforts the authors have put into making sure the model behaves as expected and matches human and non-human primate measurements. If the authors wish to go this route, they can make even more direct comparisons between the monkey M1 data and their model, including showing that the dominant neural dimensions as well as the characteristic motifs (e.g., the Churchland & Shenoy “prepare trigger execute” motif) during reaching are largely conserved across monkey and model.

We are glad the reviewer appreciates our work, and agree that we could draw more comparisons to experimental data. We have focused our efforts on two new analyses.

First, Vyas et al (*Neuron*, 2020) showed, using the same visuomotor learning task we modelled, that intracortical microstimulation during movement preparation does not affect the current trial but hinders adaptation in the subsequent trial. This is exactly what we found in our simulations when we reproduced that experiment (Figure R8A,B). In addition, we also extended that work to predict that stimulation during the “feedback period” of the reach (when afferent input reaches the brain given the delay) would similarly hinder learning in the next trial (Figure R8A,B). Second, although in our model all units received a feedback signal, we know that these connections are spatially organised in the cortex. For example, as expected from their position along the neuraxis, primary motor cortex (M1) neurons have stronger sensory responses than dorsal premotor cortex (PMd) neurons, although both exhibit strong movement related activity (Omrani et al., 2016). Accordingly, one would expect to see clear learning-related activity changes in both regions, but feedback-related activity changes should be less clear in PMd than M1. This is exactly what we found when we quantified the learning-related and feedback-related activity changes in simultaneous PMd and M1 recordings during VR adaptation from (Perich et al., 2018) (Figure R9).

We have added these new analyses as part of main Figure 5 and as new Supplementary Figures 8 and 9, and discussed them appropriately.

Minor grammar issues:

Line 111: “network’s ability.... <missing word TO> ... flexibly...”

Fixed.

Line 117: “end of the line there are two “had”

That was intentional (past perfect tense).

Line 124: “learned” not “learnt”

We wrote the manuscript in British English, where learnt is the preferred spelling.

Targeted disruption of network activity at different phases of a trial impairs learning in the next trial

Figure R8. Brief, selective manipulation of network activity during different epochs (top schematic) does not impair execution of the current trial (first and second rows), but hinders learning in the next trial (bottom row). These experiments replicate and extend the work in (Vyas *et al.*, 2020).

[figure redacted]

Figure R9. A. We analysed the activity of simultaneously recorded neural populations from M1 and PMd during the same visuomotor adaptation task we simulated here (data from Perich *et al.*, *Neuron* 2018). **B.** PMd neurons showed a large learning related (feedforward) activity change after adaptation, but the second feedback-related peak observed in M1 was largely absent. Data for one representative session. **C.** Summary results for all sessions from two different monkeys confirm the differences between M1 and PMd.

Reviewer 3

A widely accepted assumption of motor adaptation involves a combination of the "forward internal model," which predicts the sensory consequences of ongoing actions, and the "feedback-driven process" of error minimization. However, the authors of this study argue against the necessity of forward internal models, claiming that the error-minimization computation alone can accomplish motor adaptation in visuomotor rotation tasks. This indicates that predicting sensory consequences is unnecessary, and error minimization is a sufficient computation.

To support their claim, the authors used a recurrent neural network (RNN) to generate the desired movement in a center-out-reaching task and adapt like its biological counterpart when the visuomotor rotation occurs. They compared the feedforward and feedback-related internal dynamics of RNN units and biological circuits to prove that error minimization is sufficient. Their RNN suggested two critical components in motor adaptation: 1) the error between the target and actual positions as an objective function and 2) a biologically plausible learning rule. The authors then demonstrated that their RNN could explain multiple motor adaptation phenomena reported in previous literature.

While this manuscript is very interesting and raises a fundamental question about the necessity of the forward internal model, it requires improvement in multiple aspects to claim sufficiency. The following concerns are divided into major and minor aspects, but they are not prioritized.

We thank the reviewer for their appreciation of our work, and their insightful comments. We would like to clarify that we do not argue that the brain does not need a feedforward internal model, but rather that a recurrent neural circuit can deal with noisy (new Supplementary Figure 6) and delayed feedback (new Supplementary Figure 3), and achieve successful control and adaptation without an *explicit* forward model that gets updated during learning. New detailed analyses of the computations performed by the RNN during control and adaptation show that the predictive component of the feedforward internal model computation emerges naturally within the recurrent connectivity of our model (new main Figure 4, new Supplementary Figure 3). This is consistent with a recent proposal that direct policy update may underlie sensorimotor adaptation (Hadjiosif *et al.*, 2021), and inspired by behavioural studies showing that ongoing errors may act as teacher signals for trial-by-trial learning (Albert & Shadmehr, 2016), and that feedback and feedforward controller update may be more tightly linked than traditionally thought (Maeda *et al.*, 2020). We originally validated our model by showing that it could perform motor control as well as adapt to perturbations (with a biologically plausible learning rule). In our view, this is an interesting new perspective pointing to the potential implementation of the multiple computations necessary for motor control and learning by the same recurrent circuit (Gmaz *et al.*, 2024), which had not been considered previously (Figure R10). This could be achieved because in classic optimal feedback control theory the feedback controller is simply a linear computation (Todorov & Jordan, 2002) whereas a recurrent neural network can, as we show, perform much more complex computations. Based on the comments by the reviewers, we have now clarified key assumptions and features of the model throughout the manuscript, and highlighted key aspects that were not made explicit in the original manuscript.

Besides, we have now extended the validation by comparing our model against neural recordings from monkey PMd in addition to the previous data from monkey M1 (new Supplementary Figure 8), and by adding comparisons with additional behavioural and "causal" studies (updated Figure 5, new Supplementary Figure 9). We have also performed several additional simulations —outlined below— that provide further functional insight into the workings of our model.

[figure redacted]

Figure R10, reproduces Figure R4 above. Comparison between a classic model of motor adaptation, and the model proposed in this manuscript.

Major comments

1. The authors of this study delineated their logic by dichotomizing the process into error minimization versus prediction of sensory consequences and claiming the sufficiency of error minimization. To claim that error minimization is sufficient by developing a model, the authors' model may need to be as good as an alternative model that combines the prediction of sensory consequence with error minimization computation. If alternative models can explain human behavior significantly better, both qualitatively and quantitatively, the sufficiency claim may not stand.

However, the author's manuscript does not show the performance of an alternative model that includes both predictions of sensory consequence and error minimization computation. The authors only demonstrate the performance of a model where error minimization computation is removed. It is unclear whether readers should consider this error minimization removal model as a proxy for the model with "error minimization combined with the prediction of sensory consequence." If this was the authors' intention, it would be helpful to draw the linkage.

Furthermore, it is not clear where the model's prediction of sensory consequences occurs. Is it used as part of an objective function like error minimization (i.e., minimizing prediction error of sensory consequence or something similar)?

We agree with the reviewer that this was not properly explained in our original manuscript. The reason why we did not explicitly add an alternative model in the original manuscript was that we were interested in the emergent properties of the proposed RNN model. The rationale for this modelling approach is that although the training of these models is always driven by error minimization (in machine learning terms referred to as *supervised learning*) the networks can acquire various computations during training (Rajan et al., 2016; Sussillo et al., 2015; Wang et al., 2018). Thus, we focused our new analyses on whether the RNN only performs *error minimization* or whether we can also find indications that the RNN *predicts the sensory consequences of its output*. We addressed this by investigating whether a recurrent neural network could learn to perform accurate motor control in the presence of delayed and noisy feedback signals—the two main points why an explicit forward model and a state estimator were originally introduced, e.g., (Wolpert et al., 2011)—, and whether this same network can use biologically plausible error signals and plasticity rules to achieve adaptation. Our simulations show that this is indeed the case, and that such a model

recapitulates numerous observations from behavioural and electrophysiological experiments —including new results presented in revised Figure 5, and Supplementary Figures 8 and 9.

Inspired by the reviewer’s comments, we have performed several additional analyses and simulations to dissect how control and adaptation happens in the model in terms of computations. First, we show that the recurrent activity effectively allows predicting future output (Figure R11A-C), and that without recurrent connections, the model cannot correct the reaches appropriately in the case of a delayed feedback signal (Figure R12). This is consistent with the notion of the network developing a (predictive) feedforward model, as the reviewer suggests in their comments. Note however, that such a predictive model was never “hard-coded” into the model, but instead evolved from the goal of achieving error minimization with delayed feedback signals. Second, this predictive model gets updated during visuomotor adaptation through a process mediated by progressive activity changes during the early (ballistic) period of movement (Figure R11A-C,G). Third, the model can achieve successful control and adaptation even in the presence of noise at different stages (Figure R13).

Figure R11, reproduces Figure R5. Characterisation of adaptation in the network model. **A.** Accuracy of decoder trained on predicting time-lagged velocity output. **B.** Same as A but for network without recurrent connectivity. **C.** Accuracy of decoder trained on the last 100 trials of adaptation degrades when tested in earlier trials indicating a learned policy shift. Single trial accuracy (grey) and smoothed with Gaussian filter of width 10 trials (black). **D.** Average activity change of all neurons in an example network. Activity change is measured as absolute difference. **E.** Time course of average activity activity change (D) for two fixed time windows within the trial at 500ms (green) and 800ms (blue) after go. **F.** Network connectivity before adaptation for example network. Single dots represent single units and their average feedback input (x-axis) and recurrent input (y-axis) weight. **G.** Activity change to respective baseline trials of example units during adaptation. Single units switch from showing feedback-related (activity change around 800ms after go) to showing learning related (activity change around 500ms after go) activity changes.

Figure R12, partly reproduces Figure R6 above. Recurrent connectivity enables stable control with delayed feedback. Three different model variants are compared in A-F. Left (A,D), a network without any recurrent connections and with no time delay in the feedback signal. Middle (B,E), the same network without recurrent connections but with a time delayed feedback signal. Right (C,F), the default model used throughout the paper with recurrent connections and a time delayed feedback signal. **A-C.** Hand trajectories produced by the RNN. **D-F.** Hand trajectories after introducing a 30° rotation of the RNN's output, to mimic a visuomotor rotation perturbation.

Figure R13, reproduces Figure R2. Control and adaptation can be achieved with noisy signals and less localised plasticity rule. **A-C.** Learning in the presence of additive input noise. **D-F.** Learning in the presence of velocity-dependent noise applied to the output. **G-I.** Learning in the presence of additive feedback noise.

To clarify further, the loss function used during training does not include an explicit component that enforces that the network “predict the sensory consequences” of its output. Nonetheless, the data in Figure R11 suggests that the model may have learned to use a similar mechanism (mediated by the recurrent connectivity) to counteract the feedback delays (Figure R12).

We have now clarified our hypothesis and the assumptions built into the model, in addition to discussing its implications in more detail throughout the manuscript —especially in the Abstract, Introduction and Discussion. We have also added these new analyses as new main Figure 4 (discussed in the new section “Recurrent neural networks learn through functionally distinct processes that are intermixed in their implementation”), and new Supplementary Figures 3 and 6. Finally, we now also discuss another conceptual contribution of our work: that recurrently connected circuits can learn and perform feedback and feedforward control, without the need of mapping each of the underlying computations onto different brain regions or neural populations, as done in traditional accounts of motor control (Cisek & Kalaska, 2010; Shadmehr & Krakauer, 2008; Todorov & Jordan, 2002).

2. In Figure 1G, what is the objective function of the RNN model without error feedback? The method section provides an equation of the loss function for Figure 1F, but it is unclear what the objective function is for Figure 1G. Assuming the loss is the same equation except for the error signal (ϵ), the model's objective function for Figure 1G becomes "minimizing RNN unit activity, weights, and biases." This comparison seems to be between a supervised learning model (Figure 1F) and an unsupervised learning model (Figure 1G), and supervised learning with additional information would perform better than the other.

From here, a philosophical question and a practical question arise. The philosophical question is, "Does the machinery become a pure unsupervised learning machine without teacher signals once the machinery is not taking error signals into account?" The practical question is, "Can't the prediction of sensory consequence and/or any other information (e.g., a discrepancy with x , y -coordinate of the targets) be an objective function to generate training signal (besides error as teacher signal as the referred) instead of fully unsupervised?"

The models in Figure 1G and F have the same loss function, what is different is the dynamics equation; in Figure 1F the error is not provided as an input to the network, meaning that we removed the following term from the dynamical equation for $x_j(t + 1)$ described at the beginning of the Methods section:

$$\sum_i F_{ji} \epsilon_i(t - \Delta)$$

We have now added a sentence in the Methods to clarify this point.

Overall, our goal with Figure 1E-G was to make the obvious point that our model trained to perform feedback-based control does need the feedback connections in order to correct for perturbations online — i.e., it is a demonstration of necessity. Crucially, no retraining was performed: it is the same model, which we had originally trained to perform feedback-based control to minimise the instantaneous position error. In Figure 1E the model is performing an eight-target centre-out reaching task, and in Figure 1F,G the feedback was rotated by 30°, to simulate a classic visuomotor adaptation task. With feedback connections, the model could use the ongoing “somatosensory” error to correct for the wrongly aimed trajectories (Figure 1F). In contrast, when these connections were severed, it reached in the wrong direction (Figure 1G).

The reviewer is right about the philosophical implications of our work with regards to internal models. Paraphrasing, our model does learn how to use error signals to correct for perturbations, and, as part of this process, it seems to develop a feedforward model that allows it to aim movements in the right direction.

We have clarified these various issues throughout the manuscript.

3. The authors argue that their RNN exhibits a dual-rate process similar to that of humans, but their claim is currently unconvincing for two reasons. Firstly, the parameters describing the dual-rate process seem incomparable to those from a human VR adaptation study (Figure 4D). Secondly, the

authors never compared their RNN model with an alternative model (i.e., predicting sensory consequence + error minimization model).

Regarding the first point, the difference in retention factors between Af and As appears negligible in their results, unlike McDougle *et al.* Moreover, Bs generally seem larger than human results, and Bf and Bs largely overlap around 0.1-0.2 (although a statistical test is needed to confirm this). This raises two questions: 1) Would a model with a single parameter B_{avg} (a non-multiple timescale model) perform significantly worse than the dual-rate model for the network-generated trajectory (estimated by AIC or any other model comparison value)? 2) What if the model with sensory consequence prediction combined with error minimization (not shown in the manuscript) is more compatible with the dual-rate process than the model with only error minimization?

Without addressing the first question, it would be difficult to claim that the "model exhibits dual-rate like a human" since Af and As do not appear to differ, and B_{avg} could be as good as having two different Bs. If adding the feedforward internal model generates dual-rate behavior better than the error minimization model, then the authors' fundamental claim that error minimization is sufficient would not hold.

We agree with the reviewer that the dual-rate process model deserved more exploration. Motivated by this comment, we have now performed additional simulations and analyses to further characterise and understand this observation.

First, we tested whether a two-timescale model explained the learning process significantly better than a one-timescale model, as the reviewer suggests (Figure R14A,B). Intriguingly, we found that a two-timescale model fitted the data better only in a subset of networks. Given that these models only varied in parameter initialisation, we thought that this discrepancy across networks should be related to the nonlinear effects of recurrent connectivity on network activity and behaviour. To test this, we performed a second set of simulations in which we limited the recurrent connectivity across units (from 100 % connection probability to 50 %). These less recurrently connected networks were more likely to be better explained by a single timescale model than fully recurrently connected networks (compare Figure R14B and R14C), suggesting that the richer nonlinearity of more densely connected networks translated into more complex adaptation behaviour.

We have now added these new analyses and simulations as part of revised Figure 5 and new Supplementary Figure 9.

Figure R14, reproduces Figure R3. Comparison of two timescale and one timescale model fits. **A.** Example model fits for one recurrently fully connected network. **B.** Parameter fits for ten networks differing in their initial seeds colour-coded based on the best fit (one process vs. two processes). Note the mixed results in terms of best model fit. **C.** Same as (B) but for networks with 50 % recurrent connectivity. Note how in this case almost all models are best described by an order one process.

4. In a similar direction, the variance of panels in figures 2C and 2F does not seem compatible (in which the model's variance is visibly smaller). If the model with a feedforward internal model (i.e., prediction of sensory consequences + error minimization) can capture the variance pattern, would the author's sufficiency claim stand?

The reviewer is right in that the variance across learning trials is lower for the model than for the monkey data, but this is expected for a “toy model” that controls a point output based on a perfect error signal, whereas animal behaviour is inherently more variable due to a variety of factors, such as fatigue, motivation, attention, sensorimotor noise, and perhaps even the overall control objective (Todorov, 2004; Todorov & Jordan, 2002). Incorporating all these factors is out of the scope of the current study, but they are intriguing topics for future research. To being to address this point, we have now investigated the effect of three different types of noise, added to the network at different stages. We found that noise in the input signal (that signals the RNN where to reach to) resulted in the highest variance across trials, most similar to the high variance we saw in the monkey data (Figure R13).

Minor comments

1. The authors clearly summarized the input and output in figure 1B. However, I'm curious about the input and output at each time step for the RNN. Based on my understanding, the input includes velocity and error at time t , while the output is the velocity at time $t+1$. Does the RNN also compute the prediction of sensory consequences, such as error calculation, and feed it back into the network?

The input to the RNN is in fact only a 3-D constant signal (see revised Methods section “*Reaching datasets for model training and testing*”). One dimension encodes the time of the “go” signal for the ongoing trial, and the other two dimensions encode the x and y coordinates of the target. Supplementary Figure 2, which we reproduce in Figure R15, shows the input and output signals for three different phases of network training. In addition to these constant inputs, the RNN receives the delayed error feedback at $t-120$ ms (to simulate a biologically plausible delay) as a fourth input (the error calculation is not done by the RNN itself; it is given). The RNN then outputs the x and y velocity at $t+1$ based on these four inputs. There is no explicit computation enforced for “predicting the sensory consequences” although the network seems to acquire this feature following initial training, as discussed above (Figure R11).

Figure R15. Initial training protocol. The initial training of the RNN is divided into three phases. In the first phase, we kept the feedback weights fixed and at small initial values (A,D). In the second phase, we lifted that constraint, and all model parameters became plastic (B,E). In the third phase, we introduced random velocity perturbations in 75% of trials (C,F). A-C. Training loss for each of the three training phases. D-F. Input (top) and output (bottom) for an example test trial for each of the three training phases.

2. I noticed a hold signal in figure 1B, but I don't know if it continues to influence the movements once the RNN begins generating them. How do the authors negate the hold signal after the initial moment of each trial?

The hold signal is a constant which is turned to zero at the “go” signal of the respective trial. It mimics the change in visual cue (from hold to go) that monkeys and humans are subject to during this type of experiments, and is commonly used in RNN models of neuroscience tasks. *We have further clarified this in the Methods.*

3. What would be the role of random velocity bump perturbation? Why was it added to the RNN training procedure? What happens if this procedure is removed?

The random bumps were added to train the feedback connections during the later stage of training, after the RNN had learned to produce reaches (Figure R15E). However, this was not critical to the results and the network was also already able to control its output without this extra training phase.

4. How is the eligibility trace $r(t)$ calculated in line 391? Is the $r(t)$ different between the equation in line 366 and the equation in line 391?

The two are the same.

5. What is the equation of the gradient-based learning rule in contrast to the local plasticity rule? Is it identical to the models in Figures 1 and 2?

Apologies for not having indicated this clearly in the methods. For gradient-based learning, we used standard gradient descent. *We have now added a new section in the Methods, “Difference to standard gradient descent in recurrent networks,” where we address how both learning rules differ.*

6. Like the claim above, would the performance stay the same if the weight update rule includes the $F_{\text{predict_sensory}}$ matrix additively (or multiplicatively)?

We are unclear as to what the reviewer exactly means here. As can be seen from the derivation of the gradient descent rule (now added to the Methods), the weight matrix needs to be multiplicative with the error.

7. In figure 3E, the authors included a model with 20% of neurons receiving feedback to match the FB/FF ratio with macaque (as shown in figure S4B). However, the manuscript notes that 73% of biological neurons have feedback responses. If the model includes a prediction of sensory consequence component, would the discrepancy decrease when 80% of model neurons receive feedback?

We thank the reviewer for pointing this out. We have devised a new analysis where we examined whether the percentage of units that receive feedback inputs in our model is the same as the percentage of units that exhibit feedback responses, and we found that these are not the same (Figure R16). Despite all units receiving feedback input (x -axis), there is a substantial amount of them that doesn’t show a feedback response in their activity (Feedback component on y -axis is 0). Interestingly, in the case in which all units can receive feedback connections, only ~60 % of neurons exhibit feedback responses, which is qualitatively similar to the monkey data in (Cross *et al.*, 2024).

For simplicity, we have decided to focus all the main figures in the paper in the general case in which all units can potentially receive feedback —while Supplementary Figures 4, 5 and 8 repeat all key simulations and analyses for different degrees of feedback connectivity.

Figure R16, partially reproduces Figure R6. **A.** Average recurrent (blue) and feedback input (red) as well as average feedback response (black). The average feedback response is computed by taking the difference in neural activity between perturbed and unperturbed trials. **B.** Feedback component against feedback input for single neurons 620ms after GO signal. **C.** Same as (B) but 800ms after GO signal. **D.** Same as (B) but for network without any recurrent connections.

8. Is the data in figure 4B human result or RNN result? It would be helpful to appreciate the result by putting the outcome of humans and RNN side-by-side. The same applies to figure 4F.

Unfortunately, the human results are not tabularly reported in the respective publications, and the figures are copyright protected. That's why we only reported the human results in Figure 4D where we found the numbers in the paper.

9. It would be helpful to provide the rationale for the learning sequence for the 1100 divided epochs.

We initially tested different training schedules, and discovered that it is critical to have an initial phase where the model learns to perform “feedforward” reaches; this is why the first 100 epochs the feedback weights were fixed—interestingly, this may not be radically different from what animals including humans do *in utero* (Zoia et al., 2007). Then we added a later phase in which the network learned to perform feedback control using incoming feedback (the last 500 epochs without perturbations). The number of epochs for each phase was chosen after qualitative checks of the produced trajectories.

We have added this explanation in the Methods.

10. Lastly, I attempted to access the authors' GitHub repository for more model details but was unsuccessful. Could you provide access to it?

Apologies for not having made the code available before; our plan is to make it publicly available once the paper is accepted. However, we have now submitted the code along with the rebuttal so the reviewer can have a look (zipped folder `feedback-driven-plasticity-code.zip`).

Reviewer 4

The authors conducted an experiment to test their hypothesis that error signals for online feedback control and local plasticity rules are sufficient to induce trial-by-trial motor adaptation to the visuomotor rotation observed in both monkeys and humans. To this end, they developed an RNN model that employed error-based learning and online feedback control using identical error feedback signals. The simulation results showed that the RNNs achieved feedback motor control and motor learning at the same time by utilizing the same error-feedback signals. Moreover, the activity changes during learning demonstrated distinct feedback and feedforward components, consistent with the observations of monkeys' M1 neuronal activity. Finally, the RNN model reproduced the characteristic behavioural features observed in monkeys and humans. Based on these findings, the authors concluded that trial-by-trial motor adaptation can be achieved by directly updating the feedback control policy with a local plasticity rule, without the need for a forward model.

This study provides valuable insights into two longstanding questions in motor control research: whether updating the forward model is essential for motor adaptation, and how motor control and motor learning can be achieved with the same circuit. However, I believe that additional discussions and analyses of the biological plausibility and similarity to experimental data would greatly increase the significance of their model as a biological model of animal motor learning.

We thank the reviewer for their appreciation of our work, and their insightful comments. Based on the comments by the reviewers, we have clarified the key questions and assumptions throughout the revised manuscript, added more comparisons to behavioural and neural data, and included a more thorough discussion on the plausibility and realism of the model. As a result, the manuscript now has up to eight new or revised main and supplementary figures presenting new analyses and simulations.

Major comments:

1) The biological plausibility of the local plasticity rule

a. The present model assumes that the error feedback signal is provided as one of the presynaptic inputs, rather than a separate teacher signal (e.g., climbing fiber input to Purkinje cells). However, it is not entirely clear how the postsynaptic neurons would be able to distinguish the error input among the numerous other inputs and use it effectively as a teacher signal. This is an important question, as it is not clear how such a distinction could be made in a real neural network.

We thank the reviewer for raising this very important point. The model does assume that a neuron in motor cortex can distinguish a feedback input from a recurrent input. As the reviewer pointed out, in cerebellum the Purkinje cells can do that as the complex spikes from the climbing fibres are different from the parallel fibre input. However, there is evidence suggesting that this could also be the case in cortex, thanks to the large dendritic trees of cortical neurons. In particular, local axonal projections tend to arrive more proximally at the dendrites whereas feedback or top-down inputs tend to arrive more distally (Larkum, 2013). These top-down inputs can trigger nonlinear dendritic events, which strongly modulate plasticity (Magee & Johnston, 2005).

We have addressed this important point in the Discussion.

b. In addition, the model assumes that only the internal connections (W) are updated, but it is unclear how these connections can be distinguished from other (W_{in} or F) inputs projected to the same postsynaptic neuron.

Thank you for raising this interesting point. We have addressed it through new simulations and identifying additional experimental evidence that lends further support to the feasibility of our model. First, we have performed additional simulations where all weights were plastic (Figure R17). Since the main results remain qualitatively unchanged, we conclude that our findings hold beyond the original assumption of localised plasticity in our model. Second, experimental and computational work provides evidence for differential plasticity at different locations along the dendritic tree (Sjöström & Häusser, 2006). Since different sources

impinge on different places on the dendritic tree—proximal for recurrent, basal for inputs, distal for feedback, (Larkum, 2013), plasticity involving these different inputs may be similarly specific.

Figure R17, partially reproduces Figure R2. Control and adaptation can be achieved with noisy signals and less localised plasticity rule. J-L. Learning with plasticity in all incoming weights.

c. Fig. 3 suggests that error signals can be distinguished from other signals by their timing within a trial: error FB input comes later (~700ms after go cue) in the trial than other FF input (~500ms). However, it raises questions about the biological plausibility of the LTP mechanism when the error input is delayed by approximately 200 ms. This is because LTP is typically thought to occur within a more precise time window, on the order of tens of milliseconds, in the case of NMDA receptor-induced spike-timing dependent plasticity [1]. The authors could provide further clarification on this point to better explain the biophysical plausibility of their proposed mechanism.

[1] Caporale, N. & Dan, Y. Spike Timing–Dependent Plasticity: A Hebbian Learning Rule. *Annu Rev Neurosci* 31, 25–46 (2008).

Feedback inputs arrive only ~120 ms after the peak feedforward activity. What we show in Figure 3 is the activity change, which is not directly linked to the inputs. To shed light on this interesting point, we have added a new analysis that directly examines the interaction between feedback inputs and recurrent dynamics (Figure R17).

Figure R17, partially reproduces Figure R6. Recurrent connectivity enables stable control with delayed feedback. **A.** Average recurrent (blue) and feedback input (red) as well as average feedback response (black). The average feedback response is computed by taking the difference in neural activity between perturbed and unperturbed trials. **B.** Feedback component against feedback input for single neurons 620ms after GO signal. **C.** Same as (J) but 800ms after GO signal. **D.** Same as (G) but for network without any recurrent connections.

We want to note, however that our plasticity rule is instantaneous, because it depends on the correlation between the presynaptic input at time t and the feedback input at time t . As the reviewer mentions, we could have implemented an STDP-like learning window by allowing correlations to follow within 10–20 ms, but this was not necessary in our model. That said, if we had needed longer plasticity lags, computational models propose that such type of learning could be implemented mechanistically via an eligibility trace (Gerstner *et al.*, 2018), via nonlinear dendritic events such as those reported in the cortex (Schiller *et al.*, 1997), or via behavioural timescale plasticity (Bittner *et al.*, 2017).

We have added Figure R17, which presents the interaction between feedback and forward inputs as part of Supplementary Figure 3.

2) Neural basis for the feedforward motor control

The authors posit that learning induces circuits in M1 to produce feedforward motor commands, while the feedback error learning model proposed by Kawato *et al.* postulates that the inverse model is acquired in the cerebellum [2] (as cited). The cerebellum fulfills the prerequisites for error-based plasticity highlighted in my previous comment, including the presence of an independent error signal (climbing fiber input) and a relatively long LTD time window (~200 ms). As such, Kawato *et al.* suggest that the cerebellum may develop an inverse model for generating feedforward motor commands. The differences between the feedback error learning model of Kawato *et al.* and the present study's approach, as well as the potential for the cerebellum's inverse model to produce feedforward motor commands, warrant further discussion.

[2] Kawato, M. & Gomi, H. A computational model of four regions of the cerebellum based on feedback-error learning. *Biological cybernetics* 68, 95–103 (1992).

Thank you for pointing out that the central ideas in our model were not clear in the original version of the paper. We have now tried to articulate the key conceptual novelties of our paper better, which we expand on below as well as in the revised version of our paper. Briefly, these are:

- 1) We addressed the hypothesis that a recurrent circuit that can control its output to minimise an ongoing error, could use the same feedback signal to learn to adapt to a persistent perturbation. By adopting such a bottom-up approach, we were able to avoid imposing detailed computations (including a feedforward model, as in the “Kawato model”); instead we let the necessary computations emerge following initial learning (Figure R18), and devised various analyses to reverse engineer them—we have done this in the new results section “Recurrent neural networks learn through functionally distinct processes that are intermixed in their implementation,” which includes a new figure (Figure R19).
- 2) Classic models of motor control and learning map the different “computations” underlying motor adaptation (computing a feedback error, updating an inverse model, etc) in a one-to-one manner to different brain regions (e.g., (Kawato *et al.*, 1987; Shadmehr & Krakauer, 2008; Todorov & Jordan, 2002)—although more recent evidence challenges this view, e.g., (Krakauer *et al.*, 2019)). Here we show that all the necessary “computations” can emerge from the intrinsic properties of recurrent circuits that control a plant and are endowed with a local plasticity rule.

Overall, our model provides computational support and integrates recent proposals based on behavioural studies that trial-by-trial error signals could be used as teaching signals for adaptation (Albert & Shadmehr, 2016), that feedback and feedforward control are intimately related (Maeda *et al.*, 2020), and that motor adaptation can be achieved purely based on policy update (Hadjiosif *et al.*, 2021), thus contributing to an ongoing reformulation of long-held positions in the field.

To clarify the key assumptions of our model and how they relate to different models in the field, we have thoroughly revised the text in the Abstract, Introduction, and the rest of the paper. We have also created a new conceptual schematic (Figure R18, added as new Supplementary Figure 1) that contrasts the key features of our model to those of classic models in the literature.

[figure redacted]

Figure R18, reproduces Figure R4. Comparison between a classic model of motor adaptation (A), and the model proposed in this manuscript (B).

Figure R19, reproduces Figure R5. Characterisation of adaptation in the network model. **A.** Accuracy of decoder trained on predicting time-lagged velocity output. **B.** Same as A but for network without recurrent connectivity. **C.** Accuracy of decoder trained on the last 100 trials of adaptation degrades when tested in earlier trials indicating a learned policy shift. Single trial accuracy (grey) and smoothed with Gaussian filter of width 10 trials (black). **D.** Average activity change of all neurons in an example network. Activity change is measured as absolute difference. **E.** Time course of average activity change (D) for two fixed time windows within

the trial at 500ms (green) and 800ms (blue) after go. **F.** Network connectivity before adaptation for example network. Single dots represent single units and their average feedback input (x-axis) and recurrent input (y-axis) weight. **G.** Activity change to respective baseline trials of example units during adaptation. Single units switch from showing feedback-related (activity change around 800ms after go) to showing learning related (activity change around 500ms after go) activity changes.

3) Relationship with behavioural data

a. While it is impressive that the model can replicate numerous behavioural features from previous studies, further elaboration on what are the specific features of the model critical to reproducing each behavioural feature would be beneficial.

This is a great suggestion. We have now added several new analyses and simulations to understand what model features are necessary for certain behaviours, and how they relate to the observed behaviour.

First, we have extended the investigation of two scales of learning by testing whether a two-timescale model explained the learning process significantly better than a one-timescale model (Figure R20). We found that a two-timescale model fitted the data better only in a subset of networks (Figure R20B). We were intrigued by this discrepancy across models that only varied in parameter initialisation, and posited that it would be related to the nonlinear effects of recurrent connectivity on network activity and behaviour. To test this intuition, we performed a second set of simulations in which we limited the recurrent connectivity across units (from 100 % connection probability to 50 %). These less recurrently connected networks were more likely to be better explained by a single timescale model than fully recurrently connected networks (compare Figure R20B and R20C), indicating that a single timescale learning process was more likely to occur in networks with less recurrent connectivity.

Figure R20, reproduces Figure R3. Comparison of two timescale and one timescale model fits. **A.** Example model fits for one recurrently fully connected network. **B.** Parameter fits for ten networks differing in their initial seeds colour-coded based on the best fit (one process vs. two processes). Note the mixed results in terms of best model fit. **C.** Same as (B) but for networks with 50 % recurrent connectivity. Note how in this case almost all models are best described by an order one process.

Second, we have added new Figure R17, which shows how the recurrent connectivity modulates and delays the incoming feedback input to update its ongoing output. While doing so requires that the model have recurrent connections, it does not require an explicit feedforward internal model.

Third, we have added additional simulations that indicate that recurrent connectivity is necessary to achieve robust control based on a delayed feedback signal (Figure R21). While networks without recurrent connectivity could still control their output during the baseline epoch in which no perturbation was applied, in the presence of a VR perturbation the “hand trajectories” could not reach the correct targets. Thus, recurrent connections are necessary for robust output control in the presence of perturbations.

We have added these analyses as part of Figure 5 and new Supplementary Figure 3.

Figure R21, partly reproduces Figure R6. Recurrent connectivity enables stable control with delayed feedback. Three different model variants are compared in A-F. Left (A,D), a network without any recurrent connections and with no time delay in the feedback signal. Middle (B,E), the same network without recurrent connections but with a time delayed feedback signal. Right (C,F), the default model used throughout the paper with recurrent connections and a time delayed feedback signal. **A-C.** Hand trajectories produced by the RNN. **D-F.** Hand trajectories after introducing a 30° rotation of the RNN's output, to mimic a visuomotor rotation perturbation.

b. The authors suggest that the 2-state model can account for the current results; however, it is not entirely clear whether the analysis truly supports the existence of two distinct states (fast and slow states). An essential finding in the previous studies on the 2-state model is the occurrence of spontaneous recovery after de-adaptation (Smith *et al.* 2015). To lend support to the 2-state model, the authors should conduct an additional simulation to determine whether spontaneous recovery is replicated with their model.

Based on this comment and a similar remark by reviewer 1, we have now performed a direct comparison between a single-rate and a dual-rate model of learning. As discussed above, learning in fully recurrently connected networks was often best captured by a dual-rate model but that was not always the case (Figure R20). The fact that the learning dynamics of less recurrently connected networks were virtually always best captured by a single-rate model suggests that complex nonlinear recurrent dynamics give rise to the two timescales.

We have also run additional simulations to determine whether our model exhibits spontaneous recovery. As shown in Figure R22, our model does not show this phenomenon. Thus, although our model can show multiple timescales of learning, it does not really capture two distinct states as defined in psychophysics experiments in humans.

We have added a new comparison between single- and dual-rate models to the paper and rephrased — softened— the description of our findings accordingly.

Figure R22. Experimental test for “spontaneous recovery” in the RNN. **A.** Schematic from Smith *et al.* 2006 illustrating spontaneous recovery behaviour in a Multi-Rate Model. After 200 trials of adaptation to a +30° VR perturbation (grey), there are 20 trials with the opposite VR perturbation of -30° (red), and finally 100 error clamp trials where the VR perturbation is turned off. A true Multi-Rate Model can show spontaneous recovery, meaning that it jumps back to what has been learned in the longer grey phase once the error clamp trials begin. **B.** Our RNN model does not show such spontaneous recovery; instead it behaves as a single process learning model.

c. In the discussion, the authors associate each learning process (fast/slow) to two neural activity components (FB/FF). However, the association between the learning processes and neural activity components is not straightforward for me. The adaptation curve in Fig. 4C measures the take-off error, which, in principle, does not reflect the feedback response. Thus, it is unclear how the feedback neural response could influence the fast learning component. The authors should provide more detailed elaboration on the association between learning processes and neural activity components to clarify this issue.

Thank you for raising this point. We have now removed these sentences linking the fast and slow learning processes to feedback-driven and feedforward activity changes because our new analyses revealed that, although all networks show a combination of feedback-driven and feedforward components during learning (Figure R19E), some do not exhibit two learning processes as defined in (Smith *et al.*, 2006) (Figure R20B).

4) The relationship between the neural activity components of FB and FF and learning

a. If I understand correctly, it would be beneficial if the authors could provide an explanation for why the monkey M1 neural activity data were not analyzed in a similar manner to the model to elucidate the FF and FB components (i.e. B-A and C-A). As these components are expected to be present in the monkey data, their analysis would provide further insight into the neural mechanisms underlying motor learning.

We approached the analysis of activity changes in the network and the monkey data in a different way because the network simulations are essentially noise free, and we had full control of the adaptation process, including the ability to turn it on or off. This allowed us to identify the FF and FB components by simulating the exact same trials three times: one time without the perturbation (epoch A in Figure 3 of the original manuscript), another with the perturbation (epoch B in the same figure), and yet one more after adaptation to the perturbation (epoch C in the same figure). In contrast, in the monkey data we did not have access to a clean epoch B where we could look at activity under the perturbation but without any learning or large amounts of trial-to-trial variability. This is why we could not identify the FB and FF components in the monkey data, only the overall activity changes.

To strengthen our comparison to experimental recordings, we have now included a similar analysis of neural recordings from monkey dorsal premotor cortex (PMd) during the same visuomotor adaptation task (data also from (Perich *et al.*, 2018)). Our rationale for including this comparison is that while both regions show adaptation-related activity changes (Perich *et al.*, 2018), in good agreement with their position along the neuraxis, M1 is more involved in feedback-based motor control than PMd (Omran *et al.*, 2016). As such, even if we do not claim that our model simulates M1, we would still expect it to look more like M1 than PMd. A direct comparison of the activity changes in these two regions confirmed this, with PMd exhibiting considerably weaker feedback-related activity changes during learning (Figure R23).

We have included this new analysis in the manuscript as part of new Supplementary Figure 8, and explained why we could not perform the analysis identifying the FF and FB components of learning in the monkey data.

[figure redacted]

Figure R23, reproduces R9 above. A. We analysed the activity of simultaneously recorded neural populations from M1 and PMd during the same visuomotor adaptation task we simulated here (data from Perich *et al.*, *Neuron* 2018). **B.** PMd neurons showed a large learning related (feedforward) activity change after adaptation, but the second feedback-related peak observed in M1 was largely absent. Data for one representative session. **C.** Summary results for all sessions from two different monkeys confirm the differences between M1 and PMd.

b. The proposed idea is that the FB component in one trial influences the synaptic weights and leads to the appearance of the FF component in the next trial. If this is indeed the case, there should be a correlation between the FB component and the FF component of the next trial, on a trial-by-trial basis. Although the authors have demonstrated trial-by-trial learning in terms of the behavioural correlation between error size and learning size (Fig. 4A), it would be highly supportive if they could demonstrate a trial-by-trial shift from FB to FF at the level of neural activity. If demonstrating the correlation between the FF and FB components is challenging due to inherent noise in the neural signal, illustrating the adaptation curve for these components can still provide more insights."

As per the reviewer's suggestion we have now included a new analysis that shows the change in activity between each adaptation trial and the corresponding baseline trial. This allowed us to quantify the FB and FF activity changes in a trial-by-trial basis in the model (Figure R19D and E above). Interestingly, the FB component dominated early during adaptation, whereas the FF component dominated later, when the network had effectively learned to re-aim its "reaches". We further show that single units show both FB and FF components (Figure R19G), as expected from the fact that the distribution of recurrent and feedback weights is largely unimodal (Figure 19F). Overall, these analyses suggest that our model learned to counteract a persistent perturbation based on a combination of feedback and feedforward processes that are intermixed in their implementation by network units.

We have added these new analyses as part of new Figure 4, which is part of the new results section "Recurrent neural networks learn through functionally distinct processes that are intermixed in their implementation."

Minor comments:

Line 51: The authors cited Kawato's studies on feedback error learning in the context of the forward model. However, these studies mainly focus on the training of the inverse model in the cerebellum. To better support the forward model-based learning, it may be beneficial to cite references such as Shadmehr *et al.* (2010) and Flanagan *et al.* (2003), which emphasize the role of prediction in motor learning.

Shadmehr R, Smith MA, Krakauer JW (2010) Error correction, sensory prediction, and adaptation in motor control. *Annual review of neuroscience* 33:89–108.

Flanagan JR, Vetter P, Johansson RS, Wolpert DM (2003) Prediction Precedes Control in Motor Learning. *Curr Biol* 13:146–150

Thank you. We have now cited these studies in the Introduction.

Line 199-200 and the legend of Fig. 3C: it would be helpful to indicate that the activity change was measured as an absolute value.

We have added a sentence that activity change is measured as an absolute value in the caption of Figure 3 as well as in the corresponding paragraph in the Results.

Line 249: Albert et al demonstrated the temporal correlation with a time shift between the feedback response and the feedforward activity in the next trial. Did the authors observe a similar temporal correlation in the RNN and monkey M1 activity?

We experimented with several different versions of a correlation analysis between neural activity and behavioural error, but due to noise inherent to the neural data and the relatively small number of adaptation trials, we could not identify a clear association.

Fig. 4B, it would be helpful to provide an explanation as to why they used Movement error instead of the takeoff error, which was used in other sections of the study.

We have now unified the metrics with the ones we used throughout the paper, and show the take off error instead. Thanks for spotting this!

Line 392: the authors should add an explanation of why they summed every fifth time step.

We added a sentence in the Methods: “*We did this to illustrate that learning can happen on a more coarse-grained timescale than the original neural dynamics.*”

Line 401: “10/s”. Does this mean “10 cm/s”?

Yes, thanks for spotting! We corrected it.

Line 440: for the analysis of Fig. 4A, the authors correlated "the absolute value of the angular error" and "the (signed) difference in the angular error," but it is not clear why this always resulted in a positive correlation. Did the authors use the absolute value for both parameters?

Yes, we did. We have replaced the text with “the difference in the *absolute* angular error”. The correlation was indeed always positive, meaning that bigger error magnitudes led to bigger error magnitude reductions from a trial to the next.

References

- Albert, S. T., & Shadmehr, R. (2016). The Neural Feedback Response to Error As a Teaching Signal for the Motor Learning System. *Journal of Neuroscience*, *36*(17), 4832–4845. <https://doi.org/10.1523/JNEUROSCI.0159-16.2016>
- Bittner, K. C., Milstein, A. D., Grienberger, C., Romani, S., & Magee, J. C. (2017). Behavioral time scale synaptic plasticity underlies CA1 place fields. *Science*, *357*(6355), 1033–1036. <https://doi.org/10.1126/science.aan3846>
- Cisek, P., & Kalaska, J. F. (2010). Neural Mechanisms for Interacting with a World Full of Action Choices. *Annual Review of Neuroscience*, *33*(1), 269–298. <https://doi.org/10.1146/annurev.neuro.051508.135409>
- Cluff, T., & Scott, S. H. (2015). Apparent and Actual Trajectory Control Depend on the Behavioral Context in Upper Limb Motor Tasks. *Journal of Neuroscience*, *35*(36), 12465–12476. <https://doi.org/10.1523/JNEUROSCI.0902-15.2015>
- Cross, K. P., Cook, D. J., & Scott, S. H. (2024). Rapid online corrections for proprioceptive and visual perturbations recruit similar circuits in primary motor cortex. *eNeuro*. <https://doi.org/10.1523/ENEURO.0083-23.2024>
- Gerstner, W., Lehmann, M., Liakoni, V., Corneil, D., & Brea, J. (2018). Eligibility Traces and Plasticity on Behavioral Time Scales: Experimental Support of NeoHebbian Three-Factor Learning Rules. *Frontiers in Neural Circuits*, *12*. <https://www.frontiersin.org/articles/10.3389/fncir.2018.00053>

- Gmaz, J. M., Keller, J. A., Dudman, J. T., & Gallego, J. A. (2024). Integrating across behaviors and timescales to understand the neural control of movement. *Current Opinion in Neurobiology*, *85*, 102843. <https://doi.org/10.1016/j.conb.2024.102843>
- Hadjiosif, A. M., Krakauer, J. W., & Haith, A. M. (2021). Did We Get Sensorimotor Adaptation Wrong? Implicit Adaptation as Direct Policy Updating Rather than Forward-Model-Based Learning. *Journal of Neuroscience*, *41*(12), 2747–2761. <https://doi.org/10.1523/JNEUROSCI.2125-20.2021>
- Kawato, M., Furukawa, K., & Suzuki, R. (1987). A hierarchical neural-network model for control and learning of voluntary movement. *Biological Cybernetics*, *57*(3), 169–185. <https://doi.org/10.1007/BF00364149>
- Krakauer, J. W., Hadjiosif, A. M., Xu, J., Wong, A. L., & Haith, A. M. (2019). Motor Learning. *Comprehensive Physiology*, *9*(2), 613–663. <https://doi.org/10.1002/cphy.c170043>
- Larkum, M. (2013). A cellular mechanism for cortical associations: An organizing principle for the cerebral cortex. *Trends in Neurosciences*, *36*(3), 141–151. <https://doi.org/10.1016/j.tins.2012.11.006>
- Lillicrap, T. P., Santoro, A., Marris, L., Akerman, C. J., & Hinton, G. (2020). Backpropagation and the brain. *Nature Reviews Neuroscience*, *21*(6), 335–346. <https://doi.org/10.1038/s41583-020-0277-3>
- Maeda, R. S., Gribble, P. L., & Pruszynski, J. A. (2020). Learning New Feedforward Motor Commands Based on Feedback Responses. *Current Biology*, *30*(10), 1941–1948.e3. <https://doi.org/10.1016/j.cub.2020.03.005>
- Magee, J. C., & Johnston, D. (2005). Plasticity of dendritic function. *Current Opinion in Neurobiology*, *15*(3), 334–342. <https://doi.org/10.1016/j.conb.2005.05.013>
- Murray, J. M. (2019). Local online learning in recurrent networks with random feedback. *eLife*, *8*, e43299. <https://doi.org/10.7554/eLife.43299>
- Omrani, M., Murnaghan, C. D., Pruszynski, J. A., & Scott, S. H. (2016). Distributed task-specific processing of somatosensory feedback for voluntary motor control. *eLife*, *5*, e13141. <https://doi.org/10.7554/eLife.13141>
- Perich, M. G., Gallego, J. A., & Miller, L. E. (2018). A Neural Population Mechanism for Rapid Learning. *Neuron*, *100*(4), 964–976.e7. <https://doi.org/10.1016/j.neuron.2018.09.030>
- Rabe, K., Livne, O., Gizewski, E. R., Aurich, V., Beck, A., Timmann, D., & Donchin, O. (2009). Adaptation to Visuomotor Rotation and Force Field Perturbation Is Correlated to Different Brain Areas in Patients With Cerebellar Degeneration. *Journal of Neurophysiology*, *101*(4), 1961–1971. <https://doi.org/10.1152/jn.91069.2008>
- Rajan, K., Harvey, C. D., & Tank, D. W. (2016). Recurrent Network Models of Sequence Generation and Memory. *Neuron*, *90*(1), 128–142. <https://doi.org/10.1016/j.neuron.2016.02.009>
- Schiller, J., Schiller, Y., Stuart, G., & Sakmann, B. (1997). Calcium action potentials restricted to distal apical dendrites of rat neocortical pyramidal neurons. *The Journal of Physiology*, *505*(3), 605–616. <https://doi.org/10.1111/j.1469-7793.1997.605ba.x>
- Shadmehr, R., & Krakauer, J. W. (2008). A computational neuroanatomy for motor control. *Experimental Brain Research*, *185*(3), 359–381. <https://doi.org/10.1007/s00221-008-1280-5>
- Sjöström, P. J., & Häusser, M. (2006). A Cooperative Switch Determines the Sign of Synaptic Plasticity in Distal Dendrites of Neocortical Pyramidal Neurons. *Neuron*, *51*(2), 227–238. <https://doi.org/10.1016/j.neuron.2006.06.017>
- Smith, M. A., Ghazizadeh, A., & Shadmehr, R. (2006). Interacting Adaptive Processes with Different Timescales Underlie Short-Term Motor Learning. *PLOS Biology*, *4*(6), e179. <https://doi.org/10.1371/journal.pbio.0040179>
- Sussillo, D., Churchland, M. M., Kaufman, M. T., & Shenoy, K. V. (2015). A neural network that finds a naturalistic solution for the production of muscle activity. *Nature Neuroscience*, *18*(7), 1025–1033. <https://doi.org/10.1038/nn.4042>
- Todorov, E. (2004). Optimality principles in sensorimotor control. *Nature Neuroscience*, *7*(9), Article 9. <https://doi.org/10.1038/nn1309>
- Todorov, E., & Jordan, M. I. (2002). Optimal feedback control as a theory of motor coordination. *Nature Neuroscience*, *5*(11), Article 11. <https://doi.org/10.1038/nn963>
- Vyas, S., O’Shea, D. J., Ryu, S. I., & Shenoy, K. V. (2020). Causal Role of Motor Preparation during Error-Driven Learning. *Neuron*, *106*(2), 329–339.e4. <https://doi.org/10.1016/j.neuron.2020.01.019>
- Wang, J., Narain, D., Hosseini, E. A., & Jazayeri, M. (2018). Flexible timing by temporal scaling of cortical responses. *Nature Neuroscience*, *21*(1), 102–110. <https://doi.org/10.1038/s41593-017-0028-6>

Wolpert, D. M., Diedrichsen, J., & Flanagan, J. R. (2011). Principles of sensorimotor learning. *Nature Reviews Neuroscience*, *12*(12), 739–751. <https://doi.org/10.1038/nrn3112>

Zoia, S., Blason, L., D'Ottavio, G., Bulgheroni, M., Pezzetta, E., Scabar, A., & Castiello, U. (2007). Evidence of early development of action planning in the human foetus: A kinematic study. *Experimental Brain Research*, *176*(2), 217–226. <https://doi.org/10.1007/s00221-006-0607-3>

Response to reviewers' comments

Reviewer #1

The authors responded to most of my comments but I remain doubtful about the originality of the proposed learning rule and the added section, I admit, made it even more confusing.

Backpropagation is gradient descent from a loss function with a step size. Here, the learning rule differs from backpropagation because the gradient of the loss function is not used to update the weights. But in this case it corresponds to a different loss, and I was missing a theoretical argument on why it works, or how to justify it from an optimization theory perspective. Does it minimize the loss function defined on page 14? If so, does that mean it's approximately aligned with the gradient? If not, what loss function does it minimize? Or is the update rule proposed by the authors only heuristic and we should just cross our fingers and hope it works? I recommend publication provided that the authors can clarify and rework the mathematical basis of the proposed rule.

The originality of this work lies in combining a feedback-controlled neural network with a local plasticity rule that updates the recurrent weights (inspired by the work in Murray, 2019) to achieve trial-by-trial motor adaptation. Error feedback plays a dual role in our model; 1) it influences the dynamics of the network directly, and 2) during the adaptation phase, it adapts the dynamics indirectly by updating the recurrent weights based on our local plasticity rule. This approach is different from previous studies (Williams & Zipser, 1989, Murray, 2019, Bellec et al., 2020, Marschall et al., 2020), where the error feedback was not a "real" input to the network, but only a signal used to update its weights.

We have reworked the Methods section to address the reviewer's open questions and also added more citations to related work our learning rule builds upon (e.g., Murray, 2019). As the reviewer pointed out, the learning rule does minimize the loss function on page 14, which is defined as follows (note that the learning rule has additional regularization terms, which are not shown below):

$$L = \frac{1}{2BT} \sum_b \sum_t \sum_{k=x,y} \epsilon_k^2(t, b)$$

In practice, the proposed learning rule does minimize the MSE and thus approximates the gradient. We agree that a detailed comparison and analysis of its mathematical foundation would be interesting, although it is out of the scope of this paper, which focuses on motor learning.

Reviewer #2 (Remarks to the Author):

I want to thank the authors for writing this very interesting manuscript. I feel that the authors have done a very commendable job in responding to all of my previous comments. There are a number of new analyses to further explore the mechanism of their model. They have also done a lot of work on re-wording and/or clarifying various aspects of this manuscript. I feel that my most pressing concerns have now been addressed, and any further back-and-forth (between me and the authors)

would only function to quench my personal remaining curiosity rather than improve the communication of the science in a meaningful way. I want to once again thank the authors for this nice work.

We are very pleased to read your enthusiasm about our work, and would like to thank you for all the suggestions, which have helped us improve the manuscript . We are also very excited about this project and see it as a starting point for future work that will hopefully quench some of that curiosity.

Reviewer #3 (Remarks to the Author):

The reviewer is deeply grateful for the authors' diligent efforts in making the manuscript clearer and more comprehensible. The reviewer also highly values the invaluable guidance provided by the code they submitted, which significantly enhanced the understanding of the manuscript.

We are glad that the reviewer finds that the revised manuscript has substantially improved the presentation of our work, and appreciate their recommendations to that regard.

The reviewer thought they did not implement the feedforward component; however, the authors expected it to emerge, which I missed. Overall, the current version clearly demonstrates that the authors are not arguing against the necessity of a feedforward controller but are emphasizing that the recurrent connection could implicitly implement one.

Indeed, one of the key hypotheses in our work was that training a (recurrent) neural network to control and output would make a feedforward controller emerge (implicitly, paraphrasing the reviewer). We are glad this is clearer in the revised manuscript.

The reviewer has minor comments.

1. I would like to believe the result in Figure 4 indicates intermixed neurons for feedback and recurrent activity, which corresponds to Figures 4A-C. However, I could see separate clouds in the lower left corner of Figure 4D (and some in Figures S2B and C). Do the authors mind if they show some numerical results for clustering analysis in the supplementary result? Otherwise, can the readers consider it trivial noise?

Thank you for, along with reviewer 4, inviting us to reconsider that figure. We have performed a new analysis that is more tailored to the specific question of whether the feedback-related and learning components of unit activity are intermixed at the single unit level. Instead of studying the weight distribution, which only gives us indirect evidence about these two processes, we have now studied the activity produced by the network. We asked whether units that were more influenced by feedback earlier during adaptation showed larger activity changes following learning, as would be expected from our learning rule in which plasticity is driven by error feedback (and the history of

recent activity). Figure R1 shows the correlation between the average feedback component in the *first* 30 adaptation trials and the average learning component measured in the *last* 30 adaptation trials (each marker is one unit). For units showing both components, we found a significant correlation ($r=0.574$, $p<0.00001$, *scipy.stats.pearsonr*, $n=229$) between the magnitude of the feedback component early during adaptation and the learning component at the end of adaptation, which supports our claim that feedback and learning are intermixed in their implementation by single units.

Figure R1. Feedback-related and learning components in network activity are overlapping at the level of single units. Network units that show a high feedback component early during adaptation also tend to show a high learning component at the end of adaptation ($r=0.574$, $p<0.00001$, *scipy.stats.pearsonr*, $n=229$ units; measured in one example network, but the result is similar in all simulated networks). Inset on the right: Percentage of units that show feedback-related (green), learning-related (blue), or both feedback-related and learning-related (gray) activity changes during adaptation across ten networks (individual gray markers).

Further examination of these activity changes provided additional insight into how our networks achieved successful motor control and adaptation. Figure R2 shows how feedback weights changed during initial training (i.e., prior to our adaptation “experiments”). In the representative network in Figure R1, ~40% of units did not show a feedback-related component in their activity and ~18% did not show a learning component. Interestingly, having a subset of units that do not participate in “feedback processing” (i.e., that do not show a feedback-related component in their activity) seems to be actively learned during the initial training phases, since many feedback weights are pushed toward zero (Figure R2). This likely prevents oscillations in the model output, because although they are present after the initial training phase 1 (cf. Figure S2D bottom panel in the original manuscript) they are mostly absent after training phase 2 (Figure S2E bottom panel).

We have added both figures to the paper as part of Figure 4 and Supplementary Figure S2, and updated the text in the Results section.

Figure R2. Initial training leads to a group of units that do not respond to feedback. Comparison

of initial feedback input weights vs feedback input weights after phases 2 and 3 (out of 3) of initial training (A and B) shows how the feedback input weights of a subset of units get pushed close to zero (C), likely to stabilize the network output and prevent oscillations. See Methods for definition of initial training phases.

2. Among many new analyses and figures, the new supplementary figure 1 is extremely helpful in understanding the paper's core message and linkage to recent literature. Would the authors consider incorporating that somewhere in the main figure?

We are glad the reviewer finds the figure informative. We have considered adding the schematic to Figure 1 of the main paper, but decided against it to avoid side-tracking the reader away from our central objective.

3. Although the reviewer could follow the logic of the model by skimming through the well-written code, the model was not trained when I ran the attached Zipcode. The README was great, and the codebase was clear, but the VR perturbation code and automatized experiment code resulted in errors.

Thank you for pointing those errors out. Apparently Pytorch changed its API to set the device (CUDA or CPU). We have updated the code and the different scripts should run now without errors.

Besides these minor comments, the reviewer thinks clarification of key assumptions and new results support their main claim.

We appreciate the reviewer's suggestions and are glad to read their positive reaction to our work.

Reviewer #3 (Remarks on code availability):

The README file attached to the codebase helped me run the well-written code. According to the instructions, I ran the code and tested whether it generated results similar to the manuscript. Although setup-related codebases (setup_parameters.py, setup_OFC_network.py) worked fine, the VR rotation code (adaptation_learning.py) and automatized experiment code (paper.py) did not work and kept providing me NaN as the outcome.

Thank you for pointing those errors out. Apparently Pytorch changed its API to set the device (CUDA or CPU). We have updated the code and the different scripts should run now without errors.

Reviewer #4 (Remarks to the Author):

First of all, I would like to respect the author's efforts, such as adding the experiment. However, I believe that some improvements need to be considered to demonstrate the significance of the study more clearly.

We are glad that the reviewer appreciates our additions to the paper, and trust that this second round of revisions will fully address their comments.

Major comments

Comment #1 Introduction is still confusing

The introduction contains multiple topics of research mixed together, making it difficult for the reader to understand what the main issue is. I find that at least two major topics are described: the first is whether the internal forward model is necessary for motor adaptation, and the second is whether the feedforward and feedback controls are implemented independently. I recommend that the authors explicitly separate these two topics and clarify the relationship between the two topics and their relevance to the current study. Particularly confusing is that there is no “forward model” in Fig. S1, the conceptual diagram of their hypothesis. Therefore, it is still unclear how these two topics (forward model and feedforward controller (expressed as an “internal model” in Fig. S1)) are connected to the current study.

Thank you for raising this important point. *We have now more clearly separated the introduction into two parts.* The first one focuses on the question of what processes/computations underlie motor adaptation, focusing on the recent debate on inverting an updated “forward model” that predicts the consequences of one’s actions vs. direct policy update. The second part states clearly the purpose of our work: to build a “minimalistic” recurrent neural network (RNN) model that learns by directly updating the control policy that it implements, and without the need of an explicit forward model that gets updated during learning (and whose inversion mediates adaptation).

With regards to the reviewer’s point about internal models, our RNN provides evidence that in principle –as it has been suggested by recent experimental work, e.g., Hadjiosif et al 2021– the brain does not need to update a forward model that is then inverted to achieve adaptation. Instead, our RNN model achieved adaptation by directly updating the recurrent connections that map inputs into motor output based on error feedback that was subject to a biologically plausible delay (Scott 2016), that is, by e.g., updating its “control policy.” This form of adaptation replicated up to five additional observations in humans and monkeys while showing activity changes that were recapitulated in monkey M1 during the same task (data from Perich et al *Neuron* 2018). *We have clarified this in the text and the revised version of Supplementary Figure S1.*

Comment #2 Evidence for independence of neural mechanisms of feedback-based correction and adaptation is not clear

(2-1) It is stated that RNN units involved in feedback-based correction and adaptation are not separated because the distribution of the recurrent and feedback weights (W and F) showed only one cluster. However, Fig. 4D shows that there is another cluster near the zero of the horizontal axis (= weight F). It is necessary to have objective reasoning to claim that this distribution is only a “single large cluster”.

Thank you for pointing this out. *We have now replaced Figure 4D by a more specific analysis as suggested by the reviewer under point (2-3), and adapted the text in the Results section accordingly.*

(2-2) I assume that the analysis of the weights in Fig. 4D is done with a model trained only on the recurrent connection (W) (this is also not clearly written, though). However, if I understand correctly, the weight F is arbitrarily initialized in the model. Since the authors must also have models with trained input and feedback weights (W_{in} and F), I suggest examining the independence of the modules in these networks.

While the feedback weight matrix, F , is arbitrarily initialized, it is trained during the initial training phase, as described in the Methods section under “*Model training procedure*” and illustrated in Supplementary Figure S2. During the adaptation experiment only the recurrent weight matrix, W , is updated, following the biologically plausible rule described in the Methods section “*A feedback-driven plasticity rule to drive trial-by-trial learning*”.

Crucially, our main results do not change qualitatively when extending the trial-by-trial plasticity rule to update all model parameters (Supplementary Figure S6J-L). We have also repeated the new analysis suggested in point (2-3) for those networks and obtained very similar results.

(2-3) Basically, the recurrent weights (W) and feedback weights (F) represent synaptic connections and do not indicate how the units are actually activated during the task. In order to directly examine the separation of functional modules, they should examine whether units with feedback (FB) and feedforward (FF) activity components, as examined in Fig. 3C, are represented in independent clusters of neurons or not.

Thank you for, along with reviewer 3, inviting us to reconsider that figure. We have performed a new analysis that is more tailored to the specific question of whether the feedback-related and learning components of unit activity are intermixed at the single unit level. Instead of studying the weight distribution, which only gives us indirect evidence about these two processes, we have now studied the activity produced by the network. We asked whether units that were more influenced by feedback earlier during adaptation showed larger activity changes following learning, as would be expected from our learning rule in which plasticity is driven by error feedback (and the history of recent activity). Figure R3 shows the correlation between the average feedback component in the *first* 30 adaptation trials and the average learning component measured in the *last* 30 adaptation trials (each marker is one unit). For units showing both components, we found a significant correlation ($r=0.574$, $p<0.00001$, *scipy.stats.pearsonr*, $n=229$) between the magnitude of the feedback component early during adaptation and the learning component at the end of adaptation, which supports our claim that feedback and learning are intermixed in their implementation by single units.

Figure R3. Feedback-related and learning components in network activity are overlapping at the level of single units. Network units that show a high feedback component early during adaptation also tend to show a high learning component at the end of adaptation ($r=0.574$, $p<0.00001$, `scipy.stats.pearsonr`, $n=229$ units; measured in one example network, but the result is similar in all simulated networks). Inset on the right: Percentage of units that show feedback-related (green), learning-related (blue), or both feedback-related and learning-related (gray) activity changes during adaptation across ten networks (individual gray markers).

Further examination of these activity changes provided additional insight into how our networks achieved successful motor control and adaptation. Figure R4 shows how feedback weights changed during initial training (i.e., prior to our adaptation “experiments”). In the representative network shown in Figure R3, ~40% of units did not show a feedback-related component in their activity and ~18% did not show a learning component. Interestingly, having a subset of units that do not participate in “feedback processing” (i.e., that do not show a feedback-related component in their activity) seems to be actively learned during the initial training phases, since many feedback weights are pushed toward zero (Figure R4). This likely prevents oscillations in the model output, because although they are present after the initial training phase 1 (cf. Figure S2D bottom panel in the original manuscript) they are mostly absent after training phase 2 (Figure S2E bottom panel).

We have added both figures to the paper as part of Figure 4 and Supplementary Figure S2, and updated the text in the Results section.

Figure R4. Initial training leads to a group of units that do not respond to feedback. Comparison of initial feedback input weights vs feedback input weights after phases 2 and 3 (out of 3) of initial training (A and B) shows how the feedback input weights of a subset of units get pushed close to zero (C), likely to stabilize the network output and prevent oscillations. See Methods for definition of initial training phases.

(2-4) Finally, it was not shown whether neurons involved in feedback-based correction and adaptation also overlap in the monkey motor cortex. Although it is difficult to show synaptic weights (Fig. 4D) in animals, it is possible to show whether neurons with significant FB and FF activity changes (Fig. 3C) overlap or not.

We are afraid that doing this in our current monkey dataset is extremely challenging and could lead to misleading conclusions for two reasons. First, while one can quantify the percentage of neurons that respond to a discrete perturbation (e.g., many studies by Stephen Scott's lab), our task is more challenging to analyze because monkeys were not only responding to a discrete perturbation; they were searching for a solution to counter the persistent visuomotor rotation by adjusting their movements. This led to substantial inter-trial variability in the early phase of adaptation, which makes it hard to carefully dissociate the feedback-related and learning components in the activity in these trials. Second, in general, it is extremely hard to assign motor cortical neurons into discrete functional categories since they seem to live along a continuum (in fact, this is a long standing, and arguably unavoidable challenge in the field, see, e.g., Fetz 1992; Churchland & Shenoy 2007; Scott 2008). This is why we have only performed this analysis in the model.

Comment #3 The significance of the new decoding analysis is not clear.

It is not clear why the ability to decode can be evidence for an update of the control policy, rather than other possibilities (e.g., an update of the forward model). I guess that even if neural activity in the motor cortex is involved in feedback or feedforward control, it is possible that neural activity in the motor cortex correlates with current or future hand velocity, depending on the presence of noise or external perturbation.

Indeed, the decoding analysis is consistent with direct policy update, but does not show irrefutable evidence that the RNN learns via direct policy update, as the results could be partly driven by forward model update. However, we note that —borrowing nomenclature from recent work in motor learning (Hadjiosif et al 2021)— our RNN is by construction a control policy (i.e., a controller), since it uses a target cue to generate recurrent dynamics that produce the continuous output that brings the “hand” towards the target (Figure 1).

The decoder analysis that the reviewer alludes to shows that a model trained at the end of adaptation to predict the produced output based on the ongoing network dynamics becomes progressively worse as one goes back in time to the early phase of adaptation. This implies that the relationship between the recurrent network dynamics (which, by construction, determines the output) and the produced output has changed during learning as a result of the recurrent weight changes imposed by our plasticity rule.

This is also where confusion may arise, because updating a forward model that predicts the sensory consequences of the network “actions” would lead to a similar decrease in our model's predictive performance as direct control policy update, as the reviewer points out. We believe however that this is an interesting feature of our model that is posited by (optimal) feedback control theory: controllers use prediction of actions to minimize errors (e.g., Todorov 2004) and as such any good controller should learn a forward model.

We have devised a new set of simulations that show directly that our networks update their control policy during learning (Figure R5). In these simulations, networks had to adapt to a larger visuomotor rotation that matched the angular interval between two consecutive targets (45°). This allowed us to show that a cue that during the baseline period produced a reach to a specific target direction (e.g., toward the -45° target, shown in yellow), at the end of adaptation made networks “reach” toward the next target (e.g., toward the -90° target, shown in yellow). These simulations thus show directly that the networks update their control policy during adaptation.

We have revised the presentation of the decoder results and included the new simulations in Figure R5 as part of the main Figure 4.

Figure R5. Demonstration that during learning our networks update the control policy that maps input to motor output. Following adaptation to a 45° visuomotor rotation, networks aimed their reaches toward the next target given the same input (shown as matched coloured trajectories). This is most clear when examining the feedforward motor output before feedback starts to arrive following the 120 ms delay (indicated by coloured circular markers). Thus, our network’s behavior is consistent with learning via direct control policy update.

Comment #4 Results do not seem to support the dual-state model

(4-1) First, I could not see why the fast and slow parameters (A_{fast} and A_{slow} , B_{fast} and B_{slow}) existed even when the best fit was achieved with the one-state model in Fig. 5C.

We included the second order parameters even when the best fit was achieved with the one-state model to show that there are two subgroups of networks based on their learning time course that are distinguishable mostly based on B_{fast} , B_{slow} . We have now clarified this in the figure caption and the Methods.

(4-2) Generally speaking, using more model parameters should increase fitting accuracy, so it is not surprising that a dual-state model has better fitting accuracy.

We agree with the reviewer that comparing fits obtained using models with different numbers of free parameters is hard, which is why we used a statistical test that takes this into account (F -test). Besides, our results speak against more parameters always fitting the behavior better: while the learning behavior for many RNNs with 100% recurrent connectivity was better fitted by a dual-state rather than a single-state model (Figure 5C), virtually all networks with 50% recurrent connectivity were best fitted by a single-state model (Figure S9A-C). This observation rules out the possibility that the better fit to the learning behavior of fully (100%) recurrently connected networks by dual-

state models is a mere byproduct of using models with more parameters; instead, this difference seems to capture a fundamental difference in the way they learn early on during adaptation.

(4-3) More critically, if spontaneous recovery is not seen, it is reasonable to assume that there are no dual-state models. I am not pointing out this result negatively, but the limitations of this present model should also be fairly described and discussed in the manuscript (not only for responses for reviewers) because such discrepancies between the experimental data and the model provide important insights for future studies.

We have now added a sentence indicating that the model does not exhibit spontaneous recovery, as one would expect based on our model architecture.

Minor comments

Line 216: Is it a "unit" instead of a "neuron"?

Yes, corrected.

References

Bellec, Guillaume, Franz Scherr, Anand Subramoney, Elias Hajek, Darjan Salaj, Robert Legenstein, and Wolfgang Maass. A Solution to the Learning Dilemma for Recurrent Networks of Spiking Neurons. *Nature Communications* 11(1), 3625 (2020).

Churchland, M. M. & Shenoy, K. V. Temporal Complexity and Heterogeneity of Single-Neuron Activity in Premotor and Motor Cortex. *Journal of Neurophysiology* 97, 4235–4257 (2007).

Fetz, E. E. Are movement parameters recognizably coded in the activity of single neurons? *Behavioral and Brain Sciences* 15, 679–690 (1992).

Hadjiosif, A. M., Krakauer, J. W. & Haith, A. M. Did We Get Sensorimotor Adaptation Wrong? Implicit Adaptation as Direct Policy Updating Rather than Forward-Model-Based Learning. *J. Neurosci.* 41, 2747–2761 (2021).

Marschall, Owen, Kyunghyun Cho, and Cristina Savin. A Unified Framework of Online Learning Algorithms for Training Recurrent Neural Networks. *J. Mach. Learn. Res.* 21(1), (2020).

Murray, J. M. Local online learning in recurrent networks with random feedback. *eLife* 8, e43299 (2019).

Scott, S. H. Inconvenient Truths about neural processing in primary motor cortex: Neural processing in primary motor cortex. *The Journal of Physiology* 586, 1217–1224 (2008)

Scott, S. H. A Functional Taxonomy of Bottom-Up Sensory Feedback Processing for Motor Actions. *Trends in Neurosciences* 39, 512–526 (2016).

Todorov, E. Optimality principles in sensorimotor control. *Nature Neuroscience* 7, 907–915 (2004).

Response to reviewers Feulner *et al*

Williams, Ronald J., and David Zipser. A Learning Algorithm for Continually Running Fully Recurrent Neural Networks. *Neural Computation* 1(2), 270–80 (1989).